# Quantification of Motion-Induced Measurement Error on Floating Lidar Systems

Felix Kelberlau[1] and Jakob Mann[2]

[1]Fugro Norway AS, Havnegata 9, 7462 Trondheim, Norway
[2]DTU Wind, Technical University of Denmark, 4000 Roskilde, Denmark

**Correspondence:** Felix Kelberlau (f.kelberlau@fugro.com)

**Abstract.** Floating lidar systems (FLS) are widely used for offshore wind site assessment and their measurements show good agreement when compared to trusted reference sources. Though, some influence of motion on mean wind speed data from FLS has to be assumed but could not have been quantified with experimental methods yet because the involved uncertainties are larger than the expected impact of motion. This study describes the motion-induced bias on horizontal mean wind speed estimates from FLS with the help of simulations of the lidar sampling pattern of a continuous-wave (CW) velocity-azimuth display (VAD) scanning wind lidar. Analytic modelling is used to validate the simulations. It is found that the mean bias depends on amplitude and frequency of motion, the angle between motion and wind direction as well as on wind speed and strength of wind shear. The simulations are used to quantify the measurement deviation that is caused by motion for the example of the Fugro SEAWATCH Wind Lidar Buoy (SWLB) carrying a ZX 300M profiling wind lidar. The strongest bias of -0.67% of the measurement values was estimated for a test case with "strong" waves aligned with the inflow wind direction. Under "normal" wave conditions the bias is smaller. The reason for these low errors lies in a fortunate combination of the frequencies of lidar prism rotation and tilt motion.

## 1 Introduction

Commercially available profiling wind lidars are accurate instruments for measuring mean wind speed and direction onshore in non-complex terrain and offshore (Emeis et al., 2007; Smith et al., 2006; Gottschall et al., 2012). Offshore, in many cases lidars are mounted on floating platforms to avoid the costs for the construction of expensive fixed platforms. When uncorrected lidar measurements from such floating lidar systems (FLS) are compared to values from fixed lidar systems of the same type, several effects can be observed: First, wind direction estimates are influenced by the heading of the FLS (Gottschall et al., 2014). Second, measurements of second order statistics (e.g., turbulence intensity) are higher because the motion of the platform adds to the measured wind speed variance (Kelberlau et al., 2020; Gutiérrez-Antuñano et al., 2018; Désert et al., 2021). The acquired measurements of the horizontal mean wind speed, though, appear to be unbiased. In other words: The motion-induced measurement error is so small that it lies well within the overall uncertainty of experimental trial setups. A more comprehensive understanding of the potential mean wind speed measurement error is crucial for wind site assessment due to the cubic relationship between wind speed and wind turbine electricity production (Heier, 2014).

Commercial deployments of different types of FLS next to meteorological masts demonstrate good agreement with reference data (Stein et al., 2015; DNV GL, 2019). Linear regression analyses according to the Carbon Trust Roadmap (Carbon Trust, 2018) show slopes close to unity as well as offsets around zero. Furthermore, classification trials of the Fugro SEAWATCH Wind LiDAR Buoy (SWLB) showed no significant sensitivity of its measurement error to environmental variables such as wave height or buoy motion parameters. This is also reported for the Fraunhofer IWES LiDAR buoy (Wolken-Möhlmann and

Gottschall, 2020). Gottschall et al. (2017) point out that, in general, sensitivity studies show no significant influence of wave conditions on the accuracy of wind speed measurements from FLS. But they add that the motion-induced measurement error might be hidden by the larger uncertainty of the reference instruments.

Several studies investigate the error of mean wind speed measurements by FLS with computer simulations. An early example of such a study is Wolken-Möhlmann et al. (2010). They conclude that the motion-induced measurement error on mean wind

speeds is not negligible and depends on the wave heights, and the error can lead to both over- or underestimation of 10 minute averaged wind speeds. They also point out that the error is caused by rotation rather than translation. Schlipf et al. (2012) present a different simplified simulation of lidar measurements under the influence of motion. In that study the lidar is assumed to follow the wave surface and only two non-zero degrees of freedom (DoF) are considered which leads to significant deviations from the behaviour of real FLS. The simulations performed by Bischoff et al. (2015) emphasize the effect of wind shear in a

non-uniform flow field but are not realistic enough to quantify the measurement error of real FLS. The more recent study by Salcedo-Bosch et al. (2021) gives a description of measurement error caused by motion in all six degrees of freedom and finds that it depends on the initial scan phase of the velocity-azimuth display (VAD) scan. Unfortunately, they do neither calculate the error based on the assumption of randomly distributed initial scan phase angles nor include the effect of wind shear in their model. Désert et al. (2021) consider the bias on mean wind speeds in their investigation of the effects of motion on turbulence

estimates with a Doppler beam swinging wind lidar.

Mangat et al. (2014) show the influence of static tilt under consideration of realistic wind shear conditions both theoretically and experimentally. Rutherford et al. (2013) and Pitter et al. (2014) extend the same assumptions to the motion of FLS but ignore the dynamic behaviour of the lidar scanning pattern entirely and therefore oversimplify the measurement error computation of FLS compared to fixed lidar systems of the same type. Bischoff et al. (2022) present a floating lidar simulator and

50 use it to estimate mean wind speed deviations caused by lidar motion during a sea trial. The results are then compared to the measurement data.

A different approach to isolate the effect of motion in an experiment is to mount a wind lidar on a motion platform and compare the measurements to values from a closely collocated fixed lidar system of the same type. Hellevang and Reuder (2013) present their results for two different lidar types (WindCube and ZephIR) and various motion cases. The chosen motion

patterns are unfortunately not typical for FLS and the test duration of each case is so short that a quantification of the motion-induced measurement error is not possible. Tiana-Alsina et al. (2015) employ a ZephIR lidar in different scenarios but also only for short periods of time, which makes statistically relevant assessments of the small motion-induced error difficult. Also Bischoff et al. (2018) report difficulties that might be caused by the limited amount of experimental data.

The study presented here analyzes and quantifies, theoretically, the motion-induced error (i.e., bias) on FLS estimates of 10-min mean wind velocity. Computer simulations were used to imitate the measurement principle of a FLS carrying a VAD scanning profiling wind lidar, taking as a reference the ZX 300M by ZX lidars (Ledbury, United Kingdom). These computer simulations are validated by means of an analytic model. The bias was analyzed for measurements at different elevations, under varying wind shear conditions, and for multiple motion states characterized by amplitude and frequency of sinusoidal motion in all six degrees of freedom (DoF). The bias is quantified for the SWLB by Fugro (Trondheim, Norway) under "normal" and "strong" wave conditions.

Next, section 2 presents how this lidar simulator works and gives basic information about the SWLB. In section 3, we describe how motion influences the reconstructed wind vectors and resulting mean wind velocity estimates of FLS with and without consideration of wind shear. For the example of the SWLB, we define realistic test cases in section 4 and present the resulting bias. In section 5 we discuss the findings of this study. The appendix A presents the equations of the analytic model and their mathematical derivation.

## 2 Materials and Methods

### 2.1 Lidar simulator

For computations in this study we developed a lidar simulator that works as follows. In a first step, a power law wind profile is calculated from a reference wind velocity at a reference height and a wind shear coefficient according to

$$\alpha = \frac{\log(U_1/U_0)}{\log(z_1/z_0)} \tag{1}$$

where $U_1$ and $U_0$ are the mean wind velocities at elevations $z_1$ and $z_0$ above sea level.

Then a linear time vector is generated with a duration $P = 10$s and a step size of 20ms. For this time vector motion data is generated based on sine functions with given amplitude, frequency, and phase. The lidar line-of-sight (LoS) data consist of a time, prism phase, and LoS velocity vector. The time vector is identical to the time vector of motion. The vector of lidar prism phase angles consists of 10 full revolutions from 0 to $2\pi$, so that one revolution per second is simulated. This corresponds to the lidar prism rotation frequency of the ZX 300 lidar series. The prism phase angles of the first beam of each revolution are evenly distributed between 0 and $2\pi$. This simulates random first phase angles. The real azimuth and zenith angle as well as the actual measurement elevation under the influence of motion are calculated for each LoS beam from the prism phase angle, the motion data, and the configured measurement elevation. For a complete description of the vector transformations required for this procedure we refer to section 2.2 of Kelberlau et al. (2020). From this geometry information, the LoS velocities for each beam are generated by projecting the height-dependent wind velocity vector onto a unit vector pointing into the LoS direction. The resulting set of synthetic lidar data is then used to reconstruct wind vectors. Each of them is based on data of 50 samples representing one second scan time and one full prism rotation. As described for example in Kelberlau and Mann (2019) the wind vector reconstruction works by applying a least squares fit to

$$v_r = |A\cos(\theta - B) + C| \tag{2}$$

where the best fit parameters $A$, $B$ and $C$ represent the wind data according to

$$v_{hor} = A/\sin(\phi)$$
$$\Theta = B \pm 180° \qquad (3)$$
$$v_{ver} = \pm C/\cos(\phi)$$

where $v_{hor}$, $v_{ver}$, $\Theta$, $\theta$, and $\phi$ are the horizontal wind speed, vertical wind speed, wind direction, real azimuth and real zenith angle respectively. The directional ambiguity of $\pm 180°$ does not affect the horizontal wind speed $v_{hor}$ and is therefore of no concern for the analysis of the motion-induced measurement bias. Not only the phase angle of the first beam in each VAD scan cycle has a strong impact on the reconstructed wind vector but also the phase offset between lidar prism and sinusoidal motion is important. In order to remove the dependence of the reconstructed mean wind velocity on this phase offset, we run each test case twenty times, each time with a phase offset of $\frac{2\pi}{20}$ from the previous run. Ten seconds of scan time with ten different first phase angles times twenty different phase offsets of motion will lead to 200 reconstructed wind vectors. The average of all these 200 values is a good approximation of the correct bias because in a real lidar application, the first phase angle of the lidar prism will be independent of the phase of motion, and each of them occurs with equal probability. The mean bias $MB$ is then calculated by

$$MB = \frac{\frac{1}{N}\sum_{n=1}^{N} v_{hor,n}}{U} - 1 \qquad (4)$$

where $N = 200$ is the total number of wind speed values $v_{hor}$ and $U$ is the reference wind velocity.

## 2.2 SEAWATCH Wind LiDAR Buoy

The SWLB by Fugro is a discus-shaped FLS with a diameter of 2.8m and a mass of 2200kg. It carries a ZX 300M VAD scanning continuous-wave profiling wind lidar with its lidar window height approximately 1.8m above the water line. The SWLB has been deployed for commercial projects around the world. Most of the collected data is used for offshore wind site assessments where the SWLB measures parameters like mean wind speeds and directions, wave conditions, water current speeds, and atmospheric parameters like temperature and humidity. Measurement campaigns usually last around 12 months for capturing seasonal effects. The SWLB is rated Stage 3 according to the Carbon Trust Roadmap for the Commercial Acceptance of Floating LiDAR Technology (Carbon Trust, 2018), which implies that at least two different SWLB units were classified against at least two different meteorological masts and several other validation trials against trusted reference sources were successfully conducted. A picture of the SWLB is shown in Figure 1.

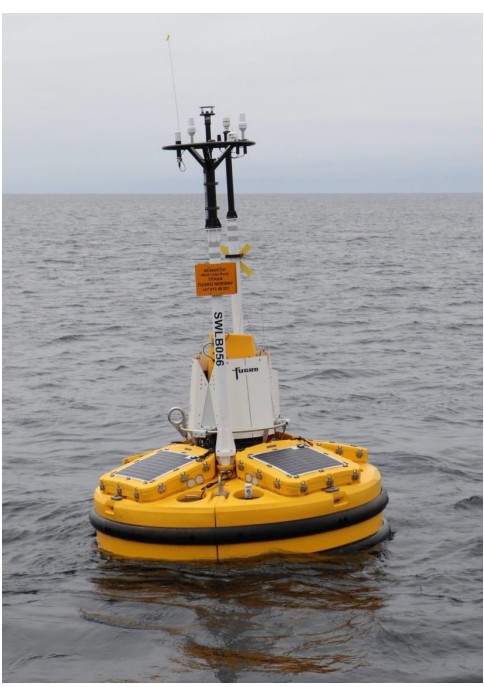

**Figure 1.** Fugro SEAWATCH Wind LiDAR Buoy

## 3  Lidar measurements under the influence of motion

### 3.1  Lidar motion in six degrees of freedom

Motion is the single characteristic that differentiates a floating from a fixed lidar system at the same location. That means systematic measurement deviations of a FLS in comparison to a fixed lidar system must be caused by its motion. We assume that this motion is restricted in two ways. First, the translational motion has to be limited to displacement around a fixed point. This assumption is true for FLS that are anchored to the seabed but violated for ship-based lidar systems. Ship-based lidar systems are therefore not covered in this study. The second assumption is that the amplitude of tilt motion of the FLS never exceeds the value at which a lidar beam is horizontal. That means for the investigated lidar system with a half-cone opening angle of $30°$, the tilt amplitude must not exceed $60°$.

The rotating prism of the lidar system serves as point of reference for the definition of six motion DoF. We define them as follows:

– Surge: Horizontal motion in mean wind direction

– Sway: Horizontal motion perpendicular to mean wind direction

– Heave: Vertical motion

- Roll: Tilt motion around surge axis (tilt leaning perpendicular to wind direction)

- Pitch: Tilt motion around sway axis (tilt leaning in wind direction)

- Yaw: Rotation around vertical axis (heading)

Motion which is not aligned with the wind direction or being perpendicular to it can be decomposed into a linear combination of its surge/roll and sway/pitch components of motion. The definition of aligning the surge direction with the wind direction in combination with the omnidirectional VAD scanning pattern of the lidar makes it therefore possible to disregard the wind direction as a parameter in this study.

## 3.2 Pitch motion with no wind shear

### 3.2.1 Low frequency pitch motion

Tilt motion, i.e., inclination of the FLS from the zenith, should be looked at separately for the pitch and roll DoF. First, we will analyse the effect of pitch motion, i.e., rotation of the FLS around a horizontal axis perpendicular to the inflow wind direction. The influence of static pitch that could be caused by steady forces as from tidal current, steady wind load, or asymmetric mass distribution on the floating platform is comparably easy to estimate. The effect of tilting a buoy in horizontal flow is identical to the effect of keeping the FLS upright in tilted flow. In this situation some of the horizontal component of the wind is interpreted as vertical inflow component. In accordance with the tilted measurement cone depicted in blue in Figure 2, the error caused by a static pitch angle $\varphi_s$ is

$$\Delta u_s = \cos \varphi_s - 1. \tag{5}$$

Assuming that the static tilt angle $\varphi_s$ will be less than a few degrees under normal operating conditions, the resulting effect of static pitch on horizontal mean wind speed measurements is low.

Larger pitch amplitudes are caused by water waves that lead to dynamic rotation of a FLS around its horizontal axes. For dynamic pitch, both the amplitude of motion $A$ and its frequency $f_p$ relative to the VAD scanning frequency $f_s$ are important ($f_s$=1 Hz in the case of the frequently used ZX 300M).

For pitch fluctuations that occur with a very low frequency $f_p \ll f_s$ the pitch angle is nearly constant during the period of each entire scanning cycle (1s) and the lidar measurement cone can be assumed frozen. A visualization is shown in Figure 2. The pitch angle alternates slowly over the course of many scan cycles between $+A$ (blue) and $-A$ (yellow). In these situations the lidar unit will measure too low horizontal velocities because according to Eq. 5, $\Delta u < 0$ for $\varphi \neq 0$. The measurement bias can be estimated by integrating Eq. 5 over half a cycle of $\varphi_s = A \sin x$

$$\Delta u = \frac{1}{\pi} \int_0^\pi \cos \left( A \sin x \right) dx - 1. \tag{6}$$

This equation can be calculated using

$$\Delta u = J_0(A) - 1 \tag{7}$$

where $J_0(A)$ is the Bessel function of the first kind and $A$ is the amplitude of the harmonic pitch oscillation. The solutions for three different pitch amplitudes $A = 5°$, $10°$, and $15°$ are visible in Figure 6 for nearly static tilt ($f_p \rightarrow 0$Hz).

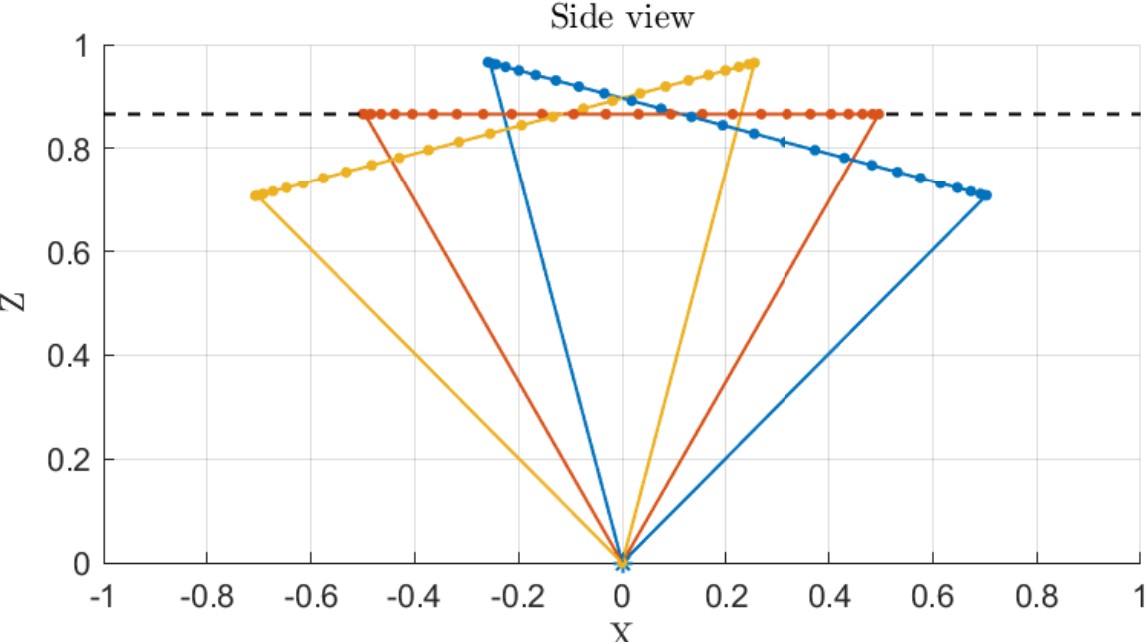

**Figure 2.** Measurement cone under influence of static or slowly changing pitch angle of $A = 20°$ with maximum positive (blue), zero (red), and maximum negative elongation (yellow). Only up- and downwind beams depicted as lines; for other beams focus locations are given by dot markers along measurement circle. Nominal measurement elevation shown (black dashed line).

### 3.2.2 Resonance frequency pitch motion

The situation is different for pitch motion that fluctuates with a frequency $f_p$ close to $f_s$. If $f_p = f_s$ the scanning "cone" for one prism rotation is no longer cone-shaped. If, e.g., the lidar beam pointing in upwind direction (i.e., the direction from where the wind blows) is pitched towards the horizon, also the lidar beam pointing in the opposite downwind direction 0.5s later will

be pitched towards the horizon. It is equally likely that these two particular beams are pitched towards the zenith or point into their unpitched direction. These three cases are visualized in Figure 3 in blue, yellow, and red respectively.

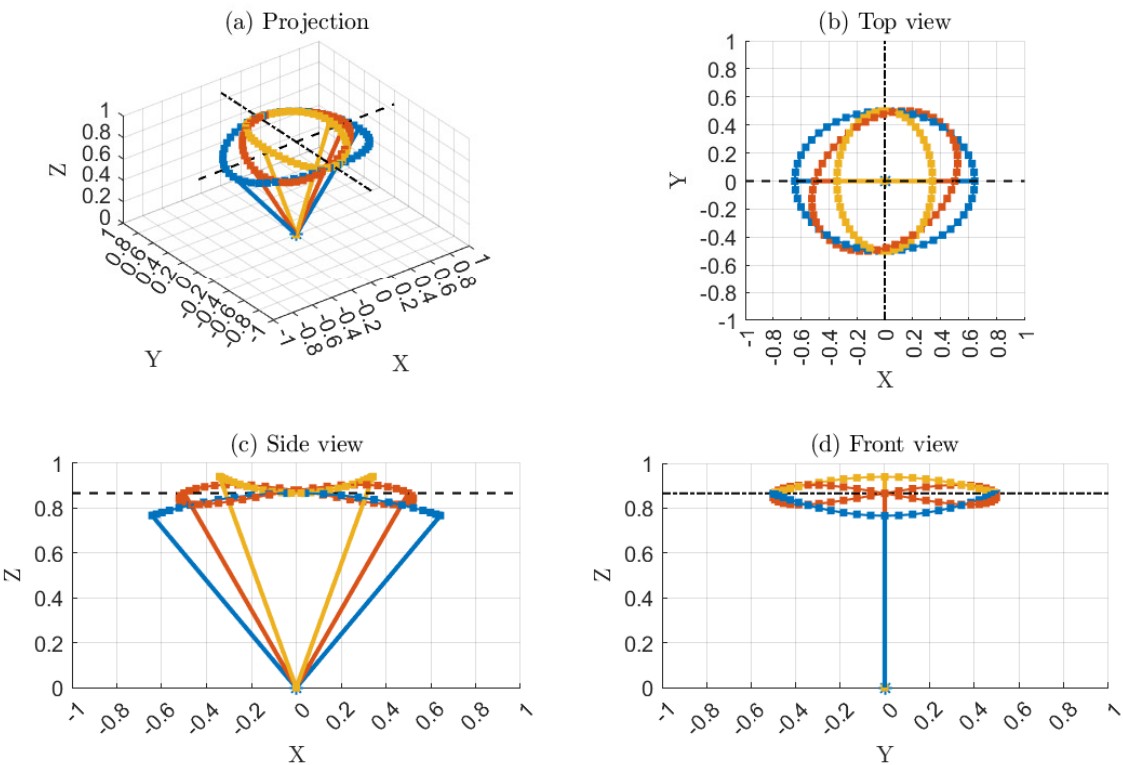

**Figure 3.** Measurement cone under influence of pitch motion with oscillation frequency $f_p = f_s = 1\text{Hz}$ and amplitude $A = 10°$. Colors represent different phase shifts between lidar prism angle and pitch motion. Only up- and downwind beams are depicted. The nominal measurement elevation is shown (black dashed line).

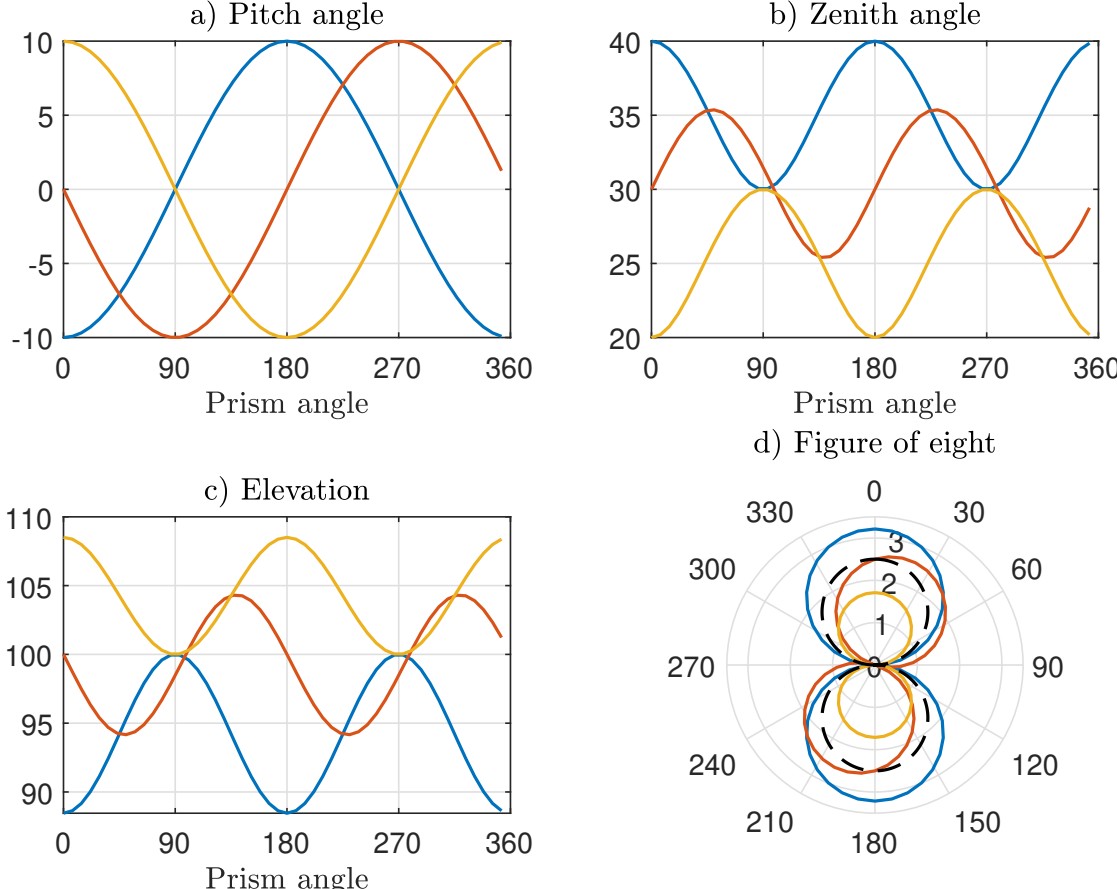

**Figure 4.** Beam geometry of VAD scanning wind lidar under influence of pitch motion with oscillation frequency $f_p = f_s = 1\mathrm{Hz}$ and amplitude $A = 10°$. The subfigures show (**a**) instantaneous pitch angles, (**b**) zenith angles, (**c**) measurement elevations, and (**d**) corresponding figures-of-eight. Colors represent different phase shifts between lidar prism angle and pitch motion corresponding to the colors in Figure 3. Figure-of-eight for fixed lidar is also shown (black dashed).The nominal measurement elevation is 100m.

Figure 4 shows plots of the most important geometrical information for these three cases. Which of them occurs depends on the phase offset between the prism angle (i.e., the lidar phase angle) and the pitch angle. The phase offset between the cases with tilt towards horizon (blue), no tilt (red), and tilt towards zenith (yellow) is 90° each. Plot a) in Figure 4 shows that if for example the beam pointing into upwind direction (blue, prism angle= 0°), is pitched by −10°, the opposing beam in downwind direction (blue, prism angle= 180°) is pitched by +10°. Plot b) shows that in this case the zenith angle, i.e., the angle between the vertical and the beam direction is 40° (blue, prism angle= 0° and 180°). This is the unpitched zenith angle plus one amplitude (the half cone opening angle $\phi = 30°$ plus 10°). Only for the beams pointing perpendicular to the

wind direction (blue, prism angle= 90° and 270°) the zenith angle equals the half cone opening angle. Plot c) shows how the measurement elevation which in this example is set to 100m is influenced by the varying zenith angles. Zenith angles of more than the half cone opening angle lead to measurements at less than 100m above ground. In this example no wind shear is assumed and therefore the measurement elevation has no influence on the LoS velocity estimates that are shown in the polar plot d). The figure-of-eight that corresponds to a fixed lidar measuring the reference wind speed is included as a dashed black line. It can be seen that the figures-of-eight representing a FLS under the influence of pitch motion in sync with the prism frequency vary substantially depending on the phase shift between motion and lidar prism angle. As expected, the wind speed estimates are significantly larger if the beams pointing in up- and downwind direction have a larger zenith angle (blue) than when their zenith angle is reduced by the pitch motion (yellow). But even in the case of zero pitch for the up- and downwind beams (red), the reconstructed wind vectors are slightly increased (and the wind direction estimate is erroneous).

These three example cases were chosen because they are particularly intuitive to understand. All other possible phase offsets between lidar prism and motion must also be considered. Figure 5 shows the relative measurement error of reconstructed wind vectors as a function of the phase offset (black dots). The three example cases are marked by vertical lines in yellow, red, and blue respectively. Since each phase offset is equally likely to occur, the expected bias for the mean wind speed is the average of all possible instantaneous measurement errors (black horizontal line). This expected bias is 1.5%. That means a FLS pitching in sync with the lidar prism frequency will overestimate the mean wind speed when no wind shear is present (and scalar averaging is used). The histogram in the lower part of the figure shows that the largest negative errors are larger ($< -0.3$) than the largest positive errors ($< 0.3$). But this effect is overcompensated by large positive errors being more frequent than large negative errors. For clarity, the histogram is based on 10,000 evenly distributed phase offset angles. A more detailed analysis would explain that the frequency-dependent positive bias seen here is entirely caused by the transversal component of the reconstructed wind vectors. Vector averaging of the wind vectors would eliminate its impact on the mean wind speed but scalar averaging, which is applied here, is affected.

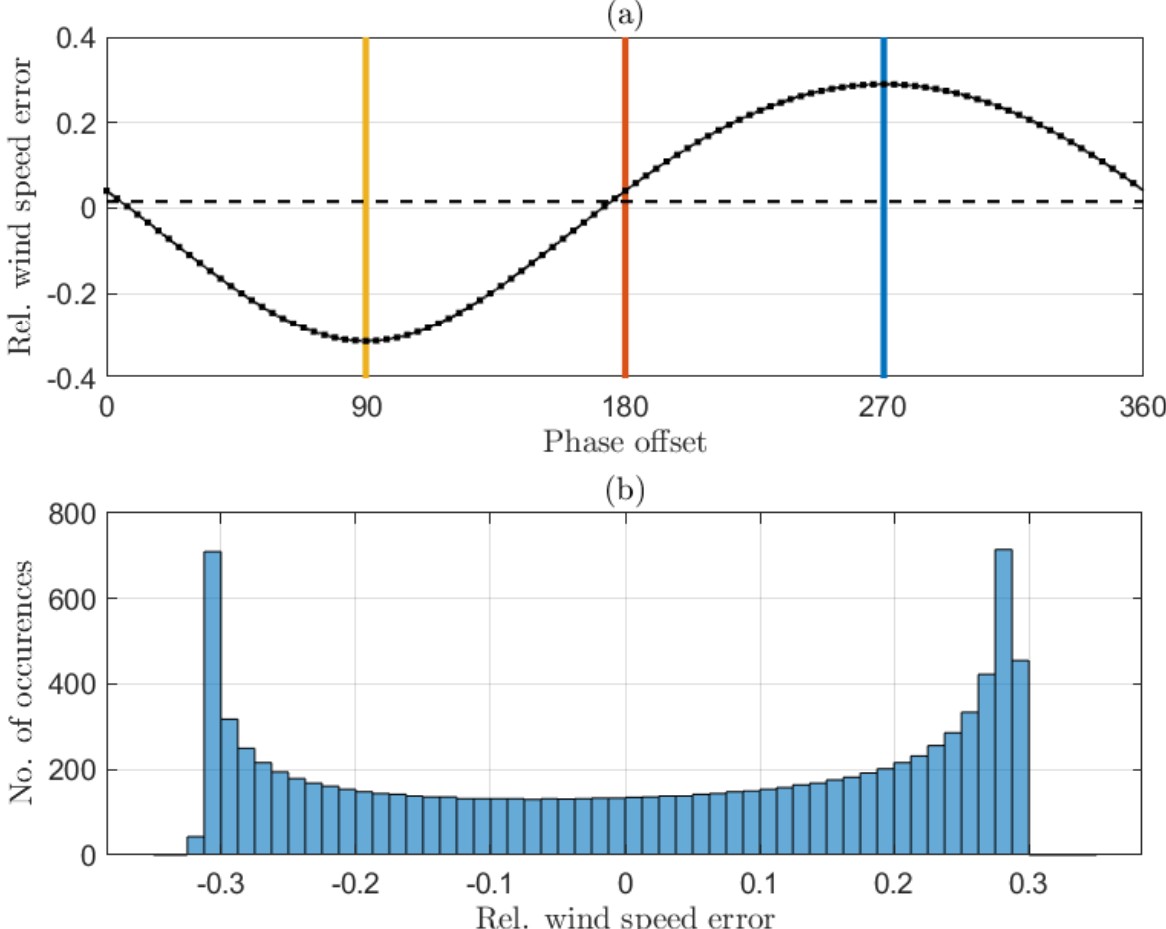

**Figure 5.** (a) Relative motion-induced measurement error caused by pitch motion with oscillation frequency $f_p = f_s = 1$Hz and amplitude $A = 10°$ as function of phase offset between lidar prism angle and motion (dot markers). Colored vertical lines mark three particular cases as in Figs. 3-4. Small non-zero bias marked by dashed horizontal line. (b) Histogram showing distribution of positive and negative wind speed errors.

### 3.2.3 Pitch motion with arbitrary frequency

In a real world application, dynamic tilt of a FLS occurs neither with very low frequency ($f_p \ll f_s$), nor with exactly the lidar prism frequency ($f_p = f_s$). The motion-induced error must therefore be determined as a function of the frequency of motion. To achieve this, we configured the simulator to estimate the motion-induced bias for three different pitch amplitudes ($5°$, $10°$, and $15°$) and for a range of motion frequencies (0 Hz... 2 Hz). Figure 6 shows the results of these computations.

It can be seen that as predicted in Eq. 7 the bias for motion with very low frequency is negative and as shown in Fig. 5 the bias at 1 Hz is positive. The largest positive biases are found at $f_p = 1$ Hz. Overall, the magnitude of the measurement bias

depends strongly on the amplitude of pitch motion. It is important to point out that in the transition from negative errors at low frequencies to positive errors at higher frequencies, there is one frequency close to 0.4 Hz at which the bias is zero. This

frequency of zero bias is independent of the motion amplitude. The second frequency of zero bias at around 1.6 Hz is of no practical relevance as such high tilt frequencies do not occur for current FLS (see section 4.1). The tilt motion frequency of a FLS is type specific and determined by its mass and hydrodynamic properties. From this visualization it is understood that for a scalar-averaging FLS with a half cone opening angle of $30°$ and a prism frequency of 1 Hz in the absence of wind shear, the tilt frequency should be close to 0.4 Hz in order to minimize its bias on mean wind speed estimates.

The overestimation of mean wind speed around $f_p = 1$ Hz is caused by the use of scalar averaging for estimating the mean wind speed. The positive bias and strong frequency dependence disappears if vector averaging of the reconstructed wind vectors is applied. This can be explained by the influence of the lateral wind speed component which increases the value of scalar averages of horizontal mean wind speed but averages out to zero for vector averages. Anyway, we recommend using scalar averaging because of its near-zero error at tilt frequencies around 0.4 Hz, which is independent of the amplitude of

motion. Scalar averaging is also the standard procedure applied by the ZX 300M lidar internally.

In Appendix A we present an analytic model for the calculation of the motion-induced error on mean wind speed estimates from a FLS. Results from this analytic solution are included in Figure 6. The purpose of comparing both methods is to validate the simulation results. Overall, the results from both methods agree well. Though, for pitch motion at higher frequencies ($> 1.4$ Hz) the results differ. While the analytic solutions converge towards $J_0(A) - 1$ the simulation results do not. This

deviation can be traced back to Eq. A8 in which we allow signed LoS velocities while in the simulator only absolute values of LoS velocities are processed (as in the ZX 300 lidar). At high frequencies of motion where the largest deviations from the ideal figure-of-eight occur, this difference has its strongest impact. At lower tilt frequencies some deviation is seen for high amplitudes of motion. This can be explained by the approximation of $A$ by means of the second order Taylor's expansion (see Eqs. A12-A13). Expanding $A$ to a higher order would probably eliminate these small deviations due to approximation. The

otherwise close agreement between the results of simulation and model support the assumptions that the simulator works well and that it can be used to predict measurements from FLS.

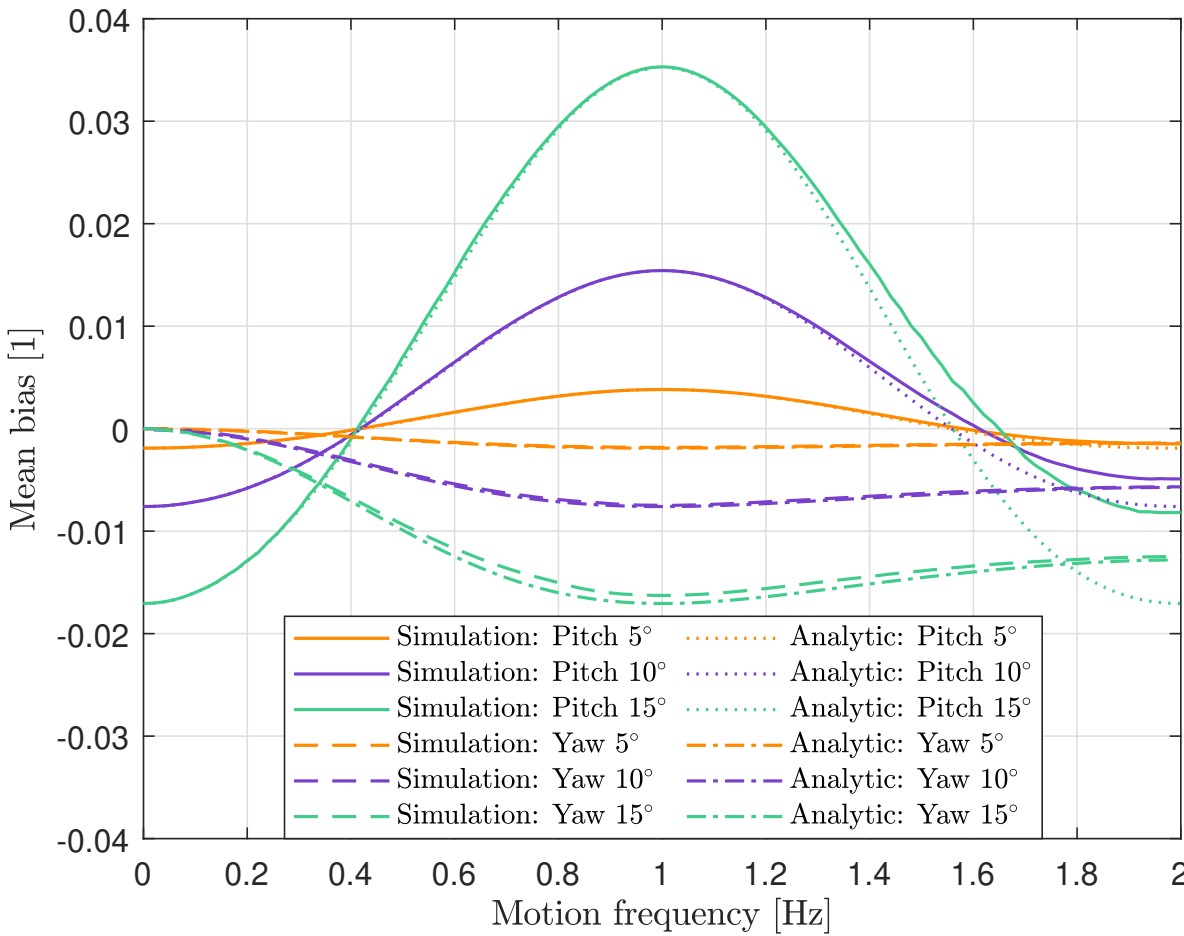

**Figure 6.** Mean bias caused by pitch (solid) and yaw (dashed) motion as function of oscillation frequency $f_p$ for three arbitrary amplitudes of motion $A = 5°, 10°,$ and $15°$ in absence of wind shear.

### 3.3 Roll motion with no wind shear

Roll motion of a FLS in the absence of wind shear is equivalent to rotating a uniform wind field around an axis parallel to the wind direction. It is therefore intuitive that in the absence of wind shear, FLS motion in roll direction has zero influence on the measurement accuracy. It is not further described here but will become relevant in section 3.6 where it is described in combination with a sheared wind field.

## 3.4 Yaw motion

Figure 6 shows the motion-induced error on estimates of horizontal mean wind speed caused by yaw motion, i.e., rotation of the FLS around the vertical axis. It can be seen that the error is zero for slow motion. This is an important finding because the restoring forces for yaw motion of a FLS are usually low which leads to low resulting motion frequencies. We will therefore disregard the effect of yawing in the following.

## 3.5 Pitch motion under the influence of wind shear

The calculations presented above in Figure 6 are based on constant wind velocities at all elevations. However, real measurements are usually influenced by a non-zero vertical wind speed gradient, i.e., wind shear (Elkinton et al., 2006). Thus, in the following, the influence of wind shear is included by introducing power law wind profiles that are characterized by the wind shear coefficient according to Eq. 1.

Figure 7 shows the motion-induced measurement error on 10-min averages of horizontal wind velocity, i.e., the mean bias $MB$, for fifteen different wind shear coefficients between 0 and 0.15. Overall, the inclusion of wind shear leads to a reduction of lidar-estimated mean wind speed. The stronger the wind shear, the stronger this effect. This is explained by the scanning geometry. According to

$$\frac{\cos(30° + \varphi) + \cos(30° - \varphi)}{2\cos(30°)} < 0 \tag{8}$$

the average measurement elevation is reduced when the pitch angle $\varphi$ is centered around zero. This can also be seen in Figure 4 (**b**) and (**c**). The reduction of measurement elevation for increased zenith angles is stronger than the increase of measurement elevation for decreased zenith angles. The on average reduced measurement elevations due to the effect of pitching lead to reduced mean wind speed estimates in the presence of wind shear profiles with lower wind speeds at lower elevations. The analytic model presented in Appendix A contains a solution for pitch and shear (A3). Its results are plotted in Figure 7 as dashed lines.

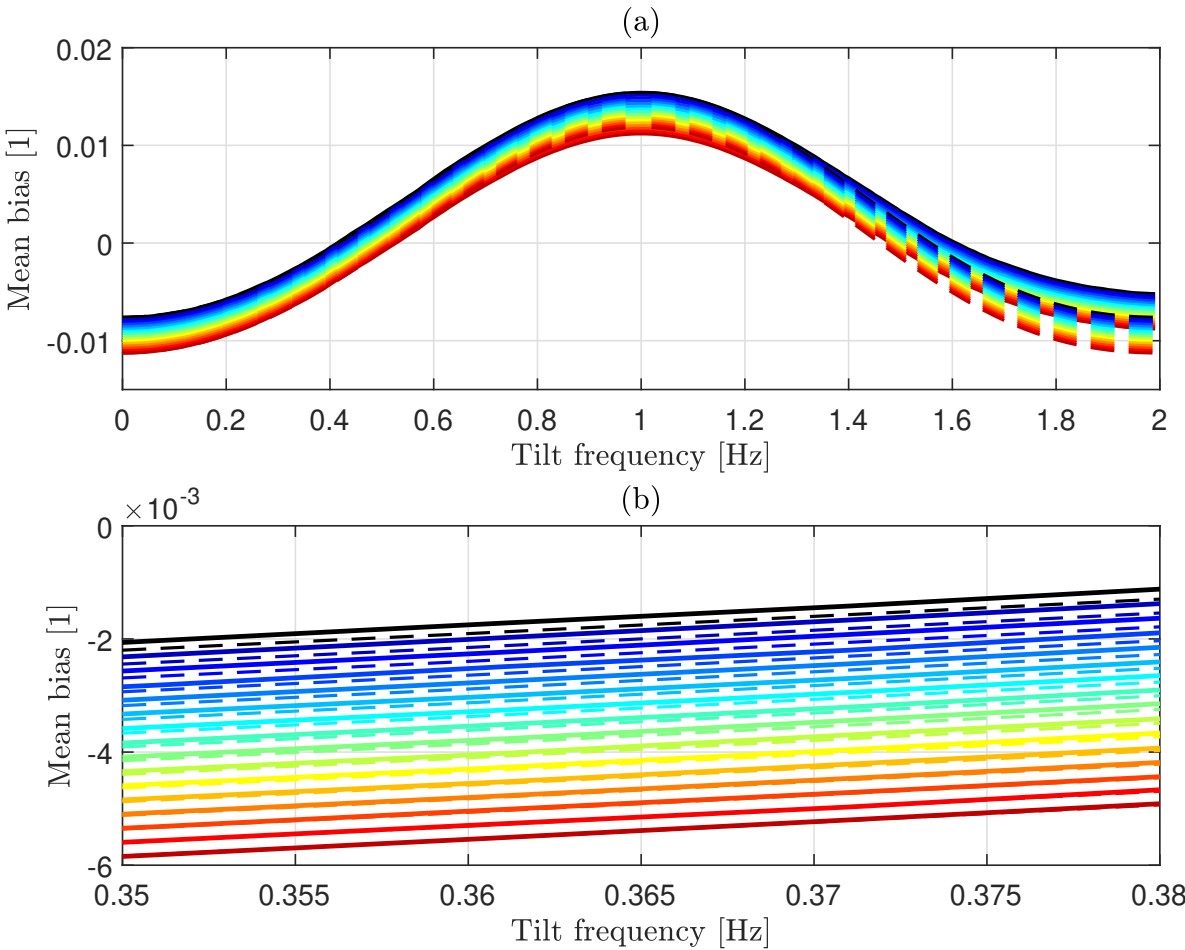

**Figure 7.** (**a**) Mean bias caused by pitch motion as function of oscillation frequency $f_p$ for amplitude of motion $A = 10°$ without (black) and with (color) consideration of wind shear characterized by shear coefficients $\alpha = 0.01$ (blue) to 0.15 (red). Results from simulation (solid) and analytic solution (dashed). (**b**) Enlarged visualization of plot above for $0.35 \leq f_p \leq 0.38$.

## 3.6 Roll motion under the influence of wind shear

Roll motion influences the elevation of the lidar beams pointing transversal to the inflow wind direction. Since the average elevation is reduced according to Eq. 8, we expect some decrease in measured mean wind speed in sheared wind fields. Figure 8 shows that the effect is independent of the motion frequency. The bias caused by roll motion is significantly lower than the effect of pitch motion. Obviously, as in the case of pitch motion, the error caused by roll motion is larger for larger wind shear coefficients. The analytic solution presented in A4 leads to the same results as the simulation.

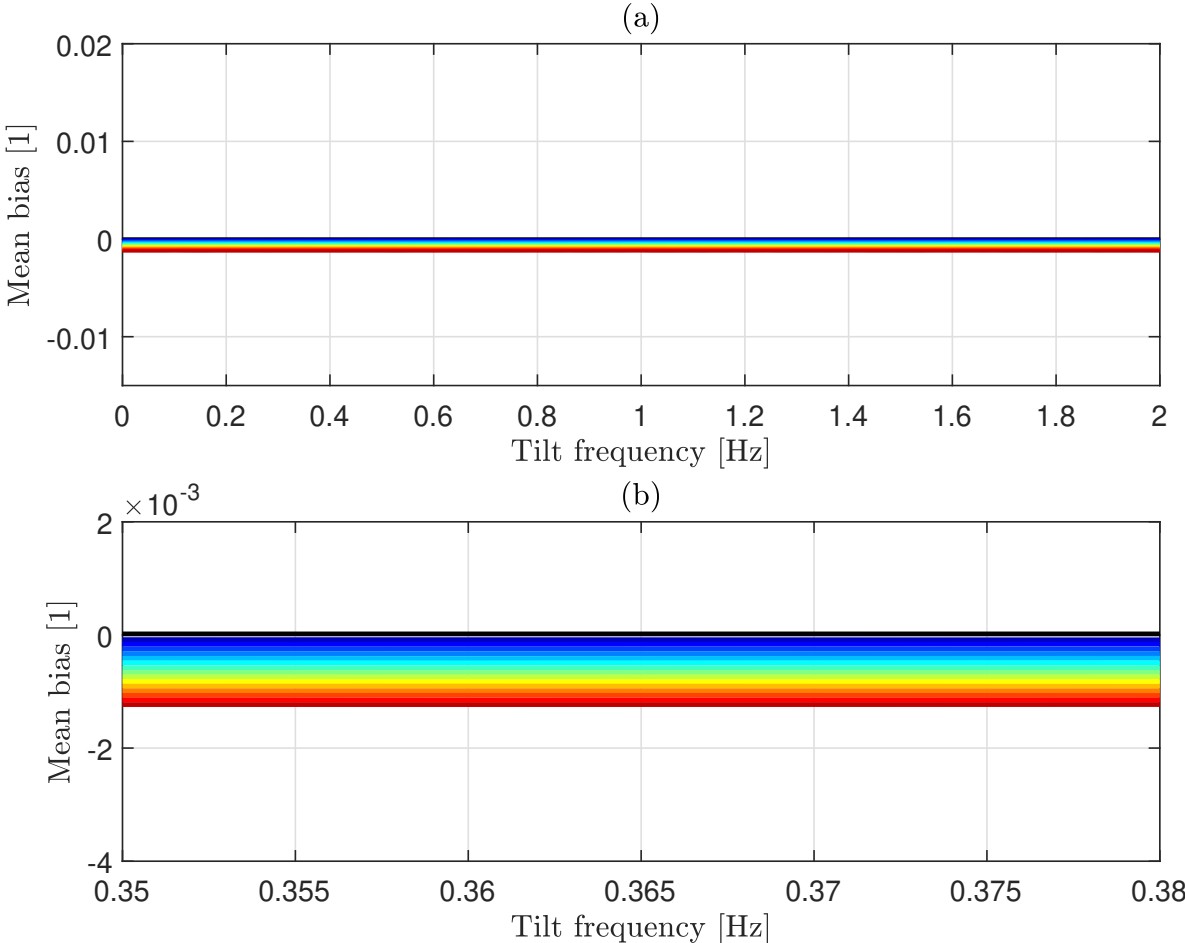

**Figure 8. (a)** Mean bias caused by roll motion as function of oscillation frequency $f_p$ for amplitude of motion $A = 10°$ without (black) and with consideration of wind shear characterized by shear coefficients $\alpha = 0.01$ (blue) to 0.15 (red). **(b)** Enlarged visualization of plot above for $0.35 \leq f_p \leq 0.38$.

## 3.7 Measurement elevation

The configuration of measurement heights of the VAD scanning profiling wind lidar influences at which focus distances from the lidar unit a FLS takes measurements. This determines the elevations above sea level at which the radial wind velocities are sampled. Tilt motion modifies the measurement elevations as shown in Figure 4 **(c)**. If the vertical gradient of horizontal mean wind speed is zero (i.e., no wind shear), the varying elevation itself has no effect on the LoS velocities. Though, in a sheared wind speed profile with usually higher wind speeds at higher elevations, the changes in elevation have an influence on the mean wind speed results as shown in Figures 7 and 8. In this study, we assume power-law wind shear profiles. For

such shear profiles it is defined that a change in vertical elevation from $z_0$ to $z_1$ by the factor $k_z = \frac{z_1}{z_0}$ leads to a change in horizontal mean wind speed from $U_0$ to $U_1$ by a factor $k_U = k_z^\alpha$. That means that the relative wind speed error caused by variations of the measurement elevations is independent of the initial elevation $z_0$. It is therefore correct that the lidar simulator computes identical relative wind speed errors for all measurement heights. For wind shear profiles that follow the power-law

the measurement error is independent of the measurement elevation. This would not be the case for other shear profiles.

## 3.8 Translational motions

Translational motion in surge, sway, and heave direction influences the lidar line-of-sight velocities as the motion vector is superimposed on the wind vector. Figure 9 shows the effect of sinusoidal oscillations in surge, sway, and heave motion with a very low frequency in (a), (b), and (c) respectively and with motion frequency equal to the lidar prism rotation frequency in

(d), (e), and (f). The upper subplots show the relative wind speed error as a function of the phase angle offset between motion and lidar prism. The sum of all possible relative wind speed error values constitutes the mean bias ($MB$) according to Eq. 4. $MB$ is shown as horizontal dashed line and its value is written in the plots. Four of the phase offsets are marked by colored vertical lines. For these phase offsets the corresponding figures-of-eight are shown in the lower subplots. These figures-of-eight are polar plots of the line-of-sight velocities as functions of the lidar prism angle. For comparison, the shape of the figure of

eight of the real wind vector without motion is shown as dashed black lines.

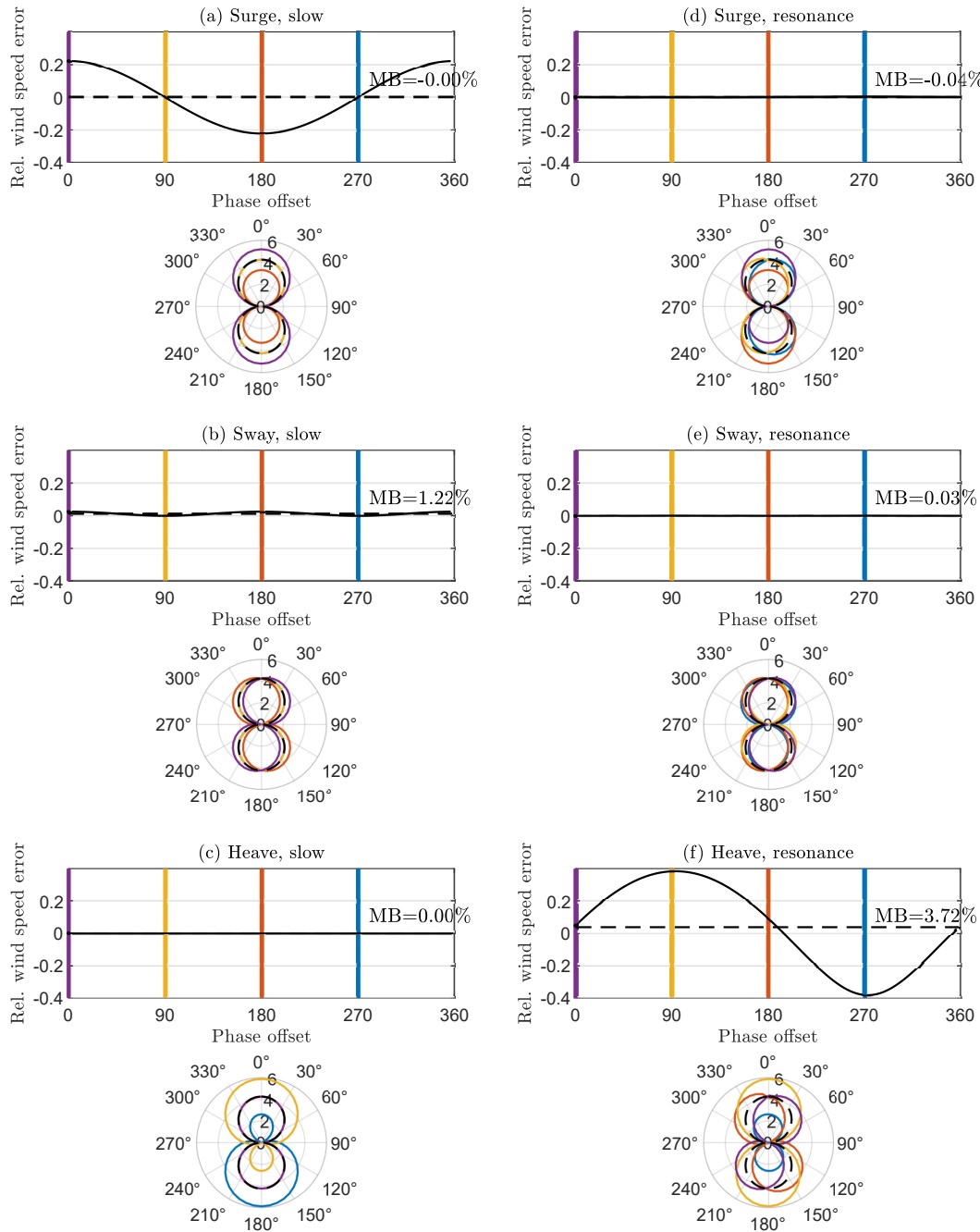

**Figure 9.** Relative motion-induced measurement error caused by motion in three translational degrees of freedom (top to bottom) with very low oscillation frequency (left) and oscillation in resonance with the lidar prism ($f_p = f_s = 1$Hz) (right). Figures of eight are given for four phase offset angles marked by colors. $U = 8.5$m/s and $\hat{v} = 1.88$m/s ($\kappa = 0.22$). Dashed black figures of eight correspond to fixed lidar measurements for comparison. Mean bias ($MB$) marked by dashed horizontal lines.

### 3.8.1 Low frequency translational motions

Figure 9 (a) shows surge motion that occurs with a very low frequency $\ll f_s$, where $f_s$ is the prism rotation frequency. It can be seen that motion into the wind direction leads to an increase in measured wind speed (purple), while motion directed in the opposite direction leads to a reduction in measured wind speed (red). The corresponding figures-of-eight vary in size depending on the phase offset. Basically, for translational motion with very low oscillation frequency the lidar estimated wind velocity vector is superimposed on the reference wind vector. For surge motion the variations average out and $MB$=0. Also in the case of heave motion with a low oscillation frequency (see Fig. 9 (c)) the mean bias of horizontal mean wind speed is zero because the resulting fluctuations of the vertical component of the lidar estimated wind vectors do not influence the horizontal wind speed. Only in sway direction slow oscillations lead to a positive measurement bias of the horizontal mean wind speed. Sway motion corresponds to a sideways component that is added to the wind blowing in surge direction. This can be seen in Figure 9 (b) where the figures-of-eight are rotated and slightly enlarged. If vector averaging would be applied, the motion-induced transversal wind velocity component would average out but since we defined $MB$ based on scalar averaging of the horizontal wind speed we are left with a positive bias.

### 3.8.2 Resonance frequency translational motions

For motion that occurs with a higher frequency the situation is different. If the frequency of translational motion is equal to $f_s$, the lidar can no longer attribute the velocity components correctly. In Figure 9 (d) this is shown for the case of surge motion that occurs in resonance with the lidar prism frequency. The horizontal wind vector component is unaffected by the horizontal motion in wind direction when the frequency of motion equals $f_s$. Instead, the motion leads to fluctuations of the vertical component of the reconstructed wind vector that does not influence $MB$. The purple figure-of-eight plot visualizes the situation in which the lidar beam that points in upwind direction samples the wind in the same moment when the lidar is moving into the wind direction with its maximum velocity. The radial velocity for the $0°$ azimuth angle is therefore increased (purple) compared to the fixed reference lidar (black, dashed). Half a second later, the lidar beam is pointing in downwind direction. At the same time the orientation of the fluctuating surge motion in resonance has changed. Now, the lidar is moving away from the wind direction with its maximum velocity. The radial velocity for the $180°$ azimuth angle is therefore decreased. The resulting figure of eight corresponds to a reconstructed wind vector with the correct horizontal wind speed and a nonzero vertical component. For other phase offsets (yellow and blue) the interpretation of the figures of eight is different but all of them lead to nearly zero error of the horizontal wind speed estimate. Sway motion in resonance with the lidar prism leads to deviations from the figure of eight of a stationary lidar but also here the horizontal wind speed component is unaffected by the motion (see Fig. 9 (e)). For heave motion in resonance this is different. Figure 9 (f) demonstrates that vertical heave motion that occurs with the prism rotation frequency leads to fluctuations of the horizontal components of the reconstructed wind vectors. As in the case of pitch motion in resonance these fluctuations do not average to zero but lead to a positive mean bias that is caused by the lateral contribution to the scalar averaged mean wind speed. A different visualization and description of the effects of translational motion on VAD scanning wind lidar is given in section 2.3.1 of Kelberlau et al. (2020).

### 3.8.3 Translational motions with arbitrary frequency

As shown above, non-zero mean bias is caused by sway motion with low oscillation frequency and by heave motion that occurs in resonance with the lidar prism rotation. The magnitude of the bias depends on the velocity of motion relative to the mean wind velocity. We therefore introduce $\kappa = \frac{\hat{v}}{U}$ where $\hat{v}$ is the peak velocity of the harmonic oscillation. Figure 10 shows the frequency dependence of $MB$ for the example of $\kappa = 0.22$. Solid curves are the simulation results and dashed curves show the corresponding analytic solutions derived in section A5 of the appendix. There is a good match between both and the only significant deviation is visible for heave at high frequencies. As for pitch motion described in section 3.2.3, these deviations can be explained by allowing signed LoS velocities in the analytical model and using only absolute values of LoS velocities in the simulator (and the ZX300 lidar).

The results show that below approximately 0.34 Hz, sway motion is the dominant contributor to $MB$. Above this value heave motion is more important for the resulting bias. The magnitude of $MB$ scales with $\kappa^2$, e.g., doubling the amplitude (and therewith $\hat{v}$) of motion while keeping the wind speed constant quadruples $MB$. As for the rotational degrees of freedom, motion in different directions can be estimated as linear combination of the involved DoF.

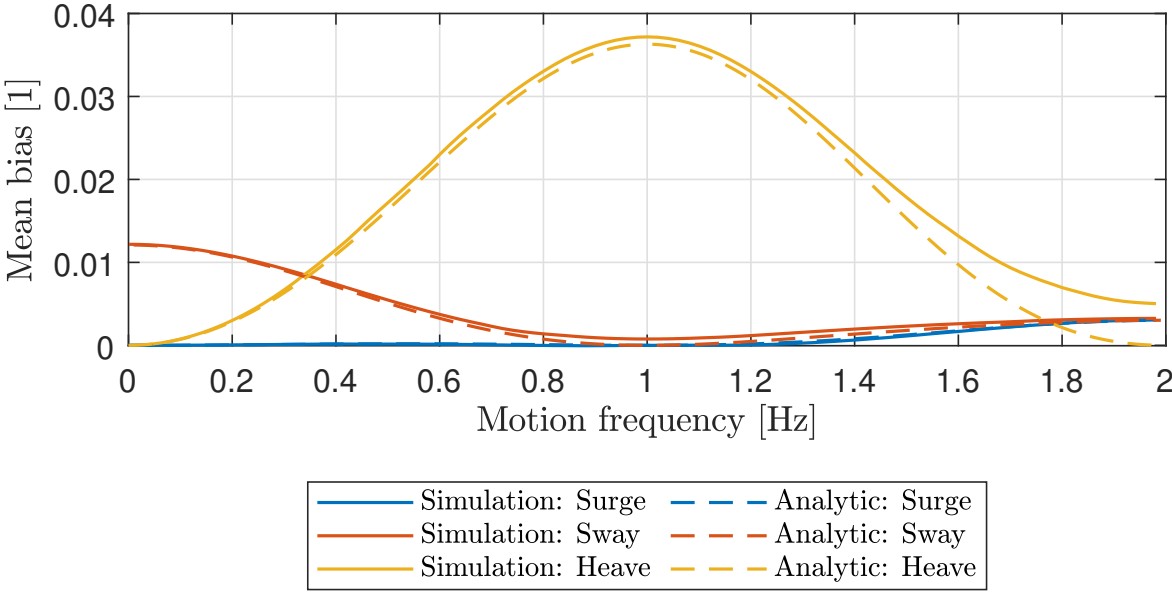

**Figure 10.** Mean bias as function of frequency of motion for translational degrees of freedom. Solid lines are results from FLS simulator and dashed lines are analytical solution presented in A5. $\kappa = 0.22$, as in Figure 9.

## 4 Quantifying the motion-induced measurement error of the SEAWATCH Wind LiDAR Buoy

Section 3 shows that the motion-induced measurement bias depends on frequency and amplitude of tilt motion, as well as the wind shear coefficient. In order to quantify the measurement error for a real FLS application we will in the following
determine realistic values for these three significant parameters. The SWLB is used in this study as an example of a frequently used commercial FLS.

### 4.1 Tilt frequency

An important driver of motion-induced measurement errors of FLS is tilt motion projected onto the mean wind direction (i.e., here pitch motion). It was found that the frequency with which pitch motion occurs influences the measurement bias.
Luckily, the tilt frequency of the SWLB is restricted to a narrow band. Tilt motion is dominated by oscillations with the natural frequency of the submerged hull that is determined by its mass and shape. The Fourier transformation of a tilt signal reveals this natural tilt frequency. Figure 11 shows the single-sided power spectrum of IMU-measured tilt motion data of SWLB unit 056 in the period from 14:00 until 16:00 UTC on 12[th] November 2021 close to the town of Titran off the coast of Frøya, Norway. The measurement data presented here is tilt in one of the buoy's local coordinate system axes. For an axis-symmetric
FLS like the SWLB the dominating tilt frequency is identical for pitch, roll, and their combination. It is therefore unnecessary to rotate the coordinate system of the motion data into a particular direction for this analysis. The red curve in the figure shows binned averages of the spectral values. The frequency bins have a width of 0.01Hz each. It can be seen that the spectrum has its maximum at 0.365 Hz which corresponds to approximately 2.74 s tilt period. The bottom plot shows an excerpt of the underlying time series of tilt signal data. The vertical lines are spaced by 2.74 s. The plot is an example of the fairly harmonic
shape of the tilt oscillations that are characteristic for the SWLB FLS type independent of varying amplitudes of motion. We will therefore set 0.365 Hz as the standard frequency for tilt motion of the SWLB.

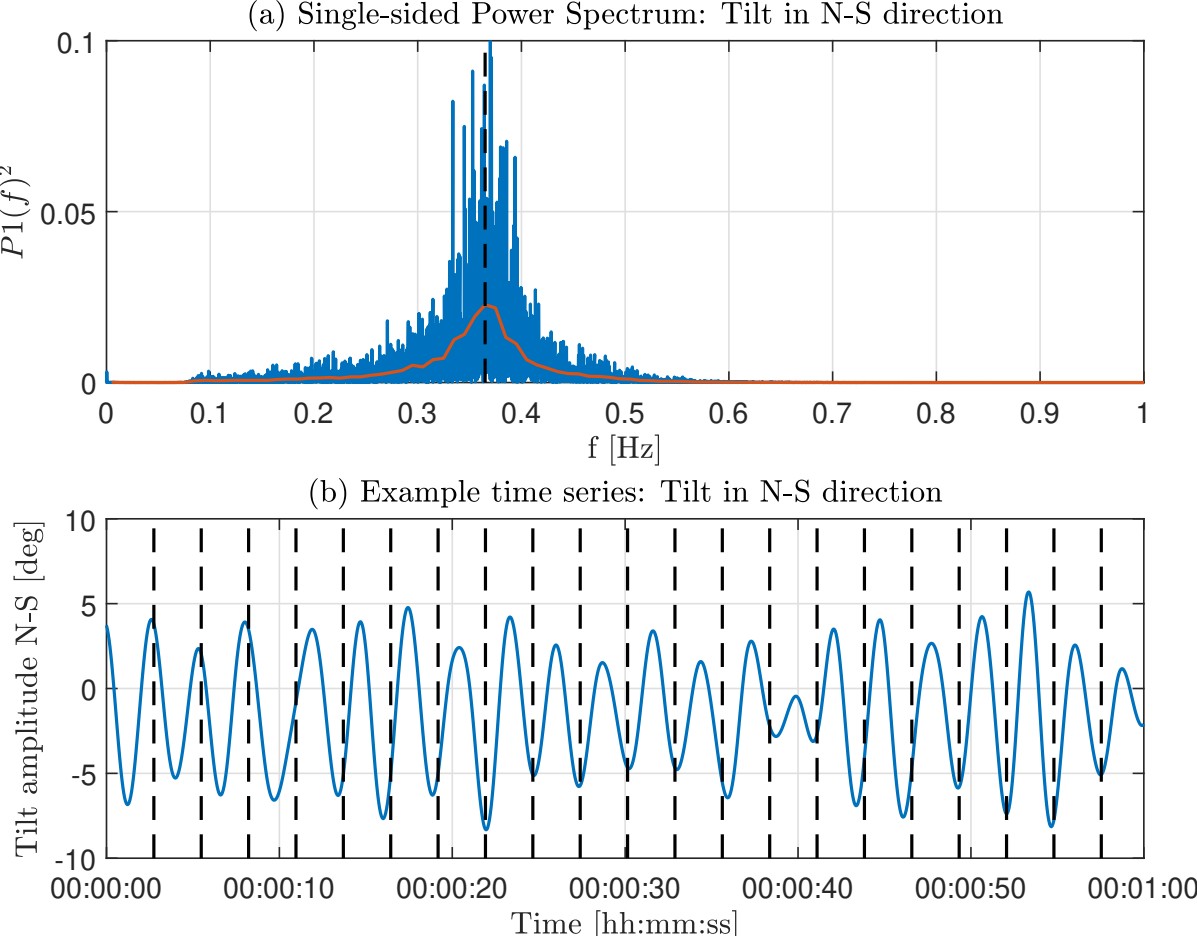

**Figure 11.** (**a**) Single-sided power spectrum of buoy tilt motion based on IMU measurement data from a SEAWATCH Wind LiDAR Buoy (blue) with bin-averaged spectral values (red). Vertical dashed line marking spectral peak at 0.365 Hz. (**b**) Excerpt of motion data in time domain with vertical dashed lines marking 2.74 s (0.365 Hz) long intervals.

## 4.2 Tilt amplitudes

The amplitude of tilt motion of a FLS depends on the prevailing sea state. For very calm seas little dynamic tilt motion is expected. By contrast, strong waves will lead to large excitation of the floating platform. The significant wave height is a well-suited parameter to describe the roughness of the sea. Significant wave height as measured by the SWLB is the average wave height, from trough to crest, of the highest third of waves within an interval of approximately 17 minutes (Sverdrup and Munk, 1947).

For determining realistic test conditions, we analyzed measurement data from three long-term measurement campaigns in the North Sea and chose an approximately 4-month long deployment from 12th March until 06th July 2016 at the East Anglia One meteorological mast (UK), that showed the highest mean of significant wave heights of four considered trials. Figure 12 shows a histogram of the observed wave heights. The mean of all significant wave heights is approximately 1.1m and the 90th percentile is found at approximately 2.0m. We will consider these wave heights the "normal" and the "strong" wave cases respectively.

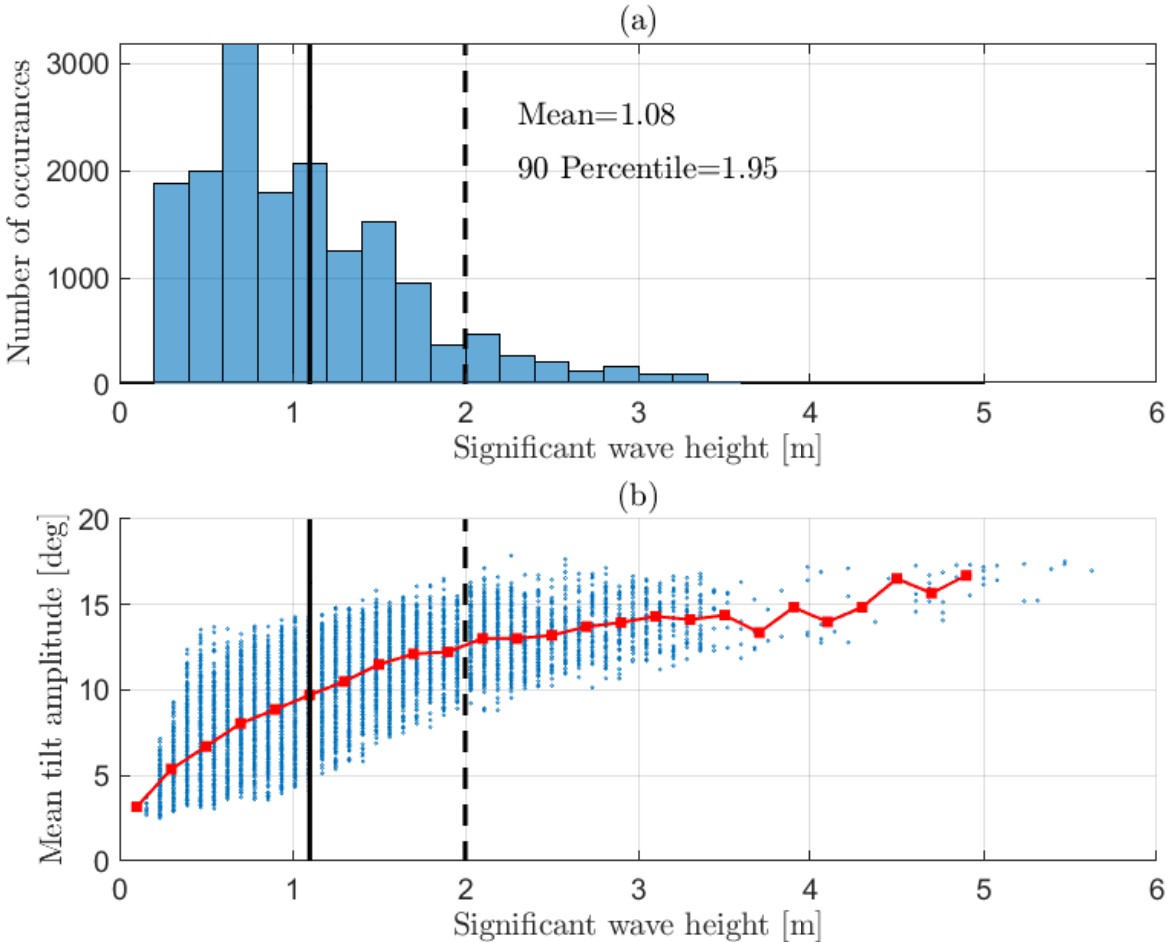

**Figure 12.** (**a**) Histogram of significant wave heights experienced during an offshore trial of the SWLB at East Anglia One met mast from March through July 2016. Mean and 90th percentile significant wave height listed in the plot. (**b**) Scatter plot of mean tilt amplitude and significant wave height (blue) including mean tilt amplitude binned by significant wave heights (red). Vertical solid and dashed lines mark the "normal" and "strong" wave cases respectively.

For a specific FLS type the significant wave height has to be transferred to a correlated amplitude of tilt motion to be able to calculate the motion-induced measurement error. This was done in the lower plot of Figure 12. The blue scatter shows data pairs of significant wave height and mean tilt amplitude of the SWLB. While the significant wave height is estimated once every 10 minutes by the FLS's internal data processing, the mean tilt amplitude is calculated for this study. The mean tilt amplitude is defined here as the average of the local maxima of the tilt time series. Here, tilt refers to the quadratic sum of the rotation angles around both horizontal axes. The red curve shows the bin-averaged relationship between significant wave height and mean tilt amplitude. It can be seen that tilt amplitudes of approximately $10°$ and $12.5°$ are expected from the "normal" and "strong" wave cases of 1.1 m and 2.0 m respectively. These tilt amplitudes can occur in any direction with regard to the mean wind direction and the relation between wind and wave direction is site-specific. In the following we will allocate the tilt amplitudes entirely to the pitch and roll DoF. If the quantification of the motion-induced error should have been performed for a certain deployment instead of a general test case, the tilt motion could have been projected onto the mean wind vector and tilt in pitch and roll directions could have been handled separately.

## 4.3  Wind shear

Usually mean wind velocities increase with vertical distance from the ground due to decreasing influence of surface roughness (Elkinton et al., 2006). The wind shear exponent $\alpha$ is calculated according to Eq. 1 from the mast-measured horizontal mean wind speeds at 80 m and 103 m above sea level.

Figure 13 is based on the same measurement data from East Anglia One that we used to determine the correlation between wave height and tilt amplitude. It shows how the significant wave height is correlated with the wind shear exponent. A trend can be seen towards stronger wind shear exponents for higher waves, likely because higher waves constitute a rougher surface for the boundary layer. According to the binned averages shown in red in the Figure, we set the wind shear exponent for "normal" waves to $\alpha = 0.08$ and for "strong" waves to $\alpha = 0.13$.

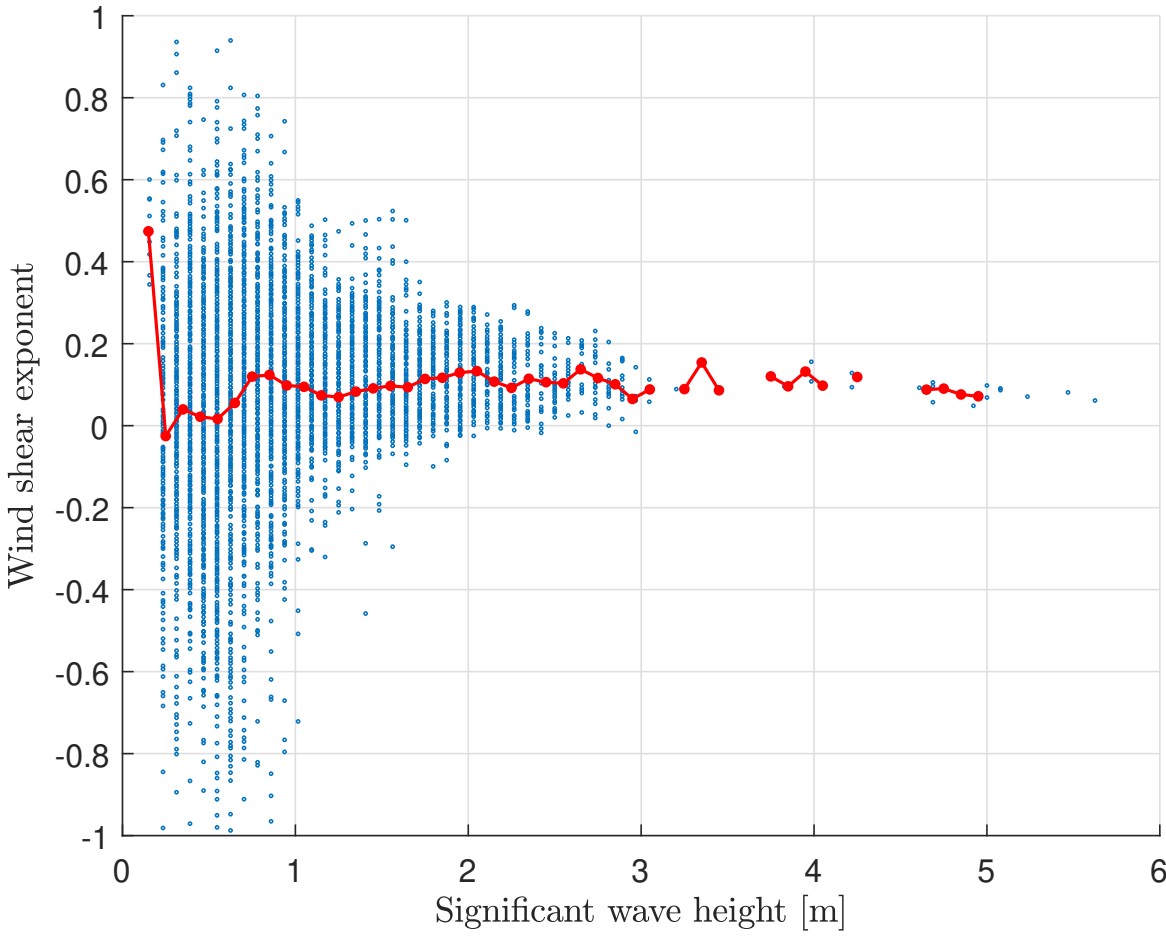

**Figure 13.** Scatter plot of wind shear exponent over significant wave height (blue) and wind shear exponent binned by significant wave height (red).

## 4.4 Translational motions

Translational motion of the lidar on the SWLB has two independent sources. The first source is rigid body motion resulting from tilt motion. The centre of rotation for pitch and roll motion lies approximately 1.3m vertically below the lidar prism. Thus, whenever the FLS exerts pitch and roll motions, also translational motions occurs, mostly in surge and sway directions. This rigid body motion occurs with the same frequency as the tilt motion that is causing it. The second source of translational movement stems from wave motion. The SWLB is considered a waverider buoy and we assume that it follows the water surface in the waves. While this is a good approximation for heave motion, the assumption is crude for horizontal translation. For calculating the mean bias of the SWLB in a realistic operational state we define circular motion with amplitudes of

**Table 1.** Summary of two test cases representing typical offshore conditions.

| Case | Tilt frequency | Tilt amplitude | Shear coefficient | Wave frequency | Amplitude | Wind speed |
|------|----------------|----------------|-------------------|----------------|-----------|------------|
| "Normal" waves | 0.365 Hz | 10.0° | 0.08 | 0.23 Hz | 0.55 m | 8.5 m/s |
| "Strong" waves | 0.365 Hz | 12.5° | 0.13 | 0.20 Hz | 1.00 m | 13.0 m/s |

$A = 0.55$ m and $1.00$ m corresponding to half of the significant wave heights determined in Figure 12 (a) for "normal" and "strong" wave cases respectively. From the experimental dataset used before we learn that such "normal" and "strong" waves typically occur with periods of $T = 4.4$ s and $5.0$ s ($0.23$ Hz and $0.20$ Hz). Such waves are in the dataset associated with wind speeds around $U = 8.5$ m/s and $13.0$ m/s respectively. With

$$\hat{v} = A \frac{2\pi}{T} \tag{9}$$

we can calculate $\kappa = 0.092$ and $0.097$ for the "normal" and "strong" wave cases.

## 4.5 Results

Table 1 summarizes the conditions that are representative for the SWLB under "normal" and "strong" wave conditions. From rotational motion we expect the largest $MB$ values if tilt motion occurs in pitch direction, i.e., in wind direction. For translational motion though we know that sway motion, i.e., motion transversal to the wind, is more important than motion in surge direction. Because we do not know which of the two types of motion is dominant we will include two orientations as test cases. First, tilt in pitch direction along with circular wave motion in surge direction and, second, tilt in roll direction along with circular wave motion in sway direction. Both combinations are computed for the "normal" and the "strong" wave case, so that a total of four test cases are defined. The $MB$ results for these four test cases are listed in Table 2. Beside of the total mean bias that considers all effects of motion some partial results are also presented. They consist of $MB$ caused by rotational motion only, rotational motion in combination with resulting rigid body motion (RBM), and translational motion only. Pitch motion leads to the highest absolute $MB$ values, especially for the larger amplitudes associated with "strong" waves ($-0.76\%$). Roll motion impacts the measurement accuracy several times less ($-0.17\%$). Adding the effect of RBM caused by the distance between the centre of rotation and the lidar prism has a very small effect on the results. Only for the case of sway motion resulting from tilt in roll direction, the effect of RBM leads to a noticeable increase of $MB$ and thereby reduces the absolute measurement bias. As shown in section 3.8, translational motion leads to overestimation of the lidar-derived mean wind speed of up to $0.26\%$ for the case of "strong" waves in sway direction. The total mean biases consist of a negative contribution from rotation and a positive contribution of translation. The largest absolute mean bias is found for "strong" waves aligned with the mean wind direction ($-0.67\%$). Most of this negative deviation is caused by pitch motion. The largest positive bias is found for "normal" waves perpendicular to the wind direction ($0.20\%$). This positive bias is dominated by the effect of translational motion.

**Table 2.** Summary of test results from simulation of mean bias ($MB$) introduced by motion of FLS

| Case | Orientation | $MB_{\text{rotation}}$ | $MB_{\text{rotation\&RBM}}$ | $MB_{\text{translation}}$ | $MB_{\text{total}}$ |
|---|---|---|---|---|---|
| "Normal" waves | Pitch/Surge | $-0.36\%$ | $-0.36\%$ | $0.07\%$ | $-0.24\%$ |
| "Strong" waves | Pitch/Surge | $-0.76\%$ | $-0.76\%$ | $0.06\%$ | $-0.67\%$ |
| "Normal" waves | Roll/Sway | $-0.07\%$ | $0.00\%$ | $0.25\%$ | $0.20\%$ |
| "Strong" waves | Roll/Sway | $-0.17\%$ | $-0.12\%$ | $0.26\%$ | $0.10\%$ |

## 5 Discussion and conclusions

Computer simulations that imitate the spatio-temporal sampling pattern of a VAD scanning FLS are performed to quantify the motion-induced error on estimates of horizontal mean wind speed. The simulation results are validated against numerical modelling of the same motion conditions. When mean wind speeds are estimated from scalar averaging, the rotational frequency of the lidar prism is an important parameter. It is set to 1 Hz which corresponds to the VAD scan frequency of the ZX 300M lidar type by ZX Lidars, UK, that is frequently used on current FLS. It is shown that also the angle between the orientation of motion and inflow wind direction is important. We defined tilt motion in wind direction as pitch motion and tilt motion perpendicular to the wind direction as roll motion.

For pitch motion, the measurement error is dependent on amplitude and frequency of motion. It is shown that FLS that oscillate with very low tilt frequencies close to 0 Hz underestimate wind speeds, while tilt frequencies close to a maximum at 1 Hz lead to overestimated horizontal wind speeds. Close to 0.4 Hz the measurement bias is approximately zero if no wind shear is assumed. The presence of positive wind shear, i.e., higher wind speeds at higher elevations, leads to a reduction of the FLS estimates of mean wind speed. For the roll DoF, the motion-induced mean bias is zero in the absence of wind shear. With wind shear it is negative and its magnitude depends on tilt amplitude and wind shear coefficient. We estimated the error caused by yaw motion to be negligible because its frequency is low for usual FLS.

In addition to the measurement error caused by rotational motion also the effect of translational motion must be considered. We defined surge and sway to be translational motion in wind direction and perpendicular to it, respectively. Surge motion has no significant influence on the mean bias of FLS but sway and heave motion increase the mean bias. The magnitude of influence of translational motion depends on its oscillation frequency relative to the lidar prism frequency as well as its peak velocity relative to the wind speed. Periodic heave motion increases the mean wind speed estimates the most when it occurs in sync with the lidar prism frequency and the impact of sway motion is the strongest when its oscillation frequency is very low.

With the aim of quantifying the motion-induced error for a real FLS, we used experimental data from a Fugro SEAWATCH Wind Lidar Buoy to determine test cases (see section 4). The cases of "normal" and "strong" wind and wave conditions were defined based on one particular deployment on the North Sea where "normal" and "strong" conditions represent the median and 90[th] percentile of significant wave heights. The hydrodymamic properties of the SWLB lead to a dominant tilt frequency

of 0.365 Hz which turns out to be fortunate for mean wind speed measurements with the ZX 300M used on the buoy as it is close to around 0.4 Hz where the bias introduced by pitch motion is zero. The resulting measurement biases for the "normal" and "strong" test cases are $-0.24\%$ and $-0.67\%$ respectively if wind and wave directions are aligned. If wind and waves occur with perpendicular directions, the respective mean bias values are $0.20\%$ and $0.10\%$. These biases are smaller than typical uncertainties of around 2% that are usually found when FLS are validated against meteorological masts. It is therefore difficult to confirm the simulation results during field campaigns. Analyses of classification trials according to Annex L of the IEC 61400-12-1 standard however have shown that the sensitivity of measurement error of the SWLB to motion and sea-state parameters is insignificant (reference on request). The simulations presented here give a complete explanation for why the motion-induced error is so small.

In this study we investigated the mean measurement error or systematic bias that motion introduces into FLS measurements according to Eq. 4. This systematic mean bias appears to be the most important error parameter because it has the potential to influence the slopes of linear regression lines as used when, e.g., the assessment procedures described in the Carbon Trust Roadmap (Carbon Trust, 2018) are followed. It would have been possible for particular test cases based on data from the simulations to also analyze the random error caused by motion, i.e., the variance of the deviations between FLS estimates of 10-minute mean wind speed and the true values. But we did not do it because the random error is strongly dependent on the number of reconstructed wind vectors per averaging interval. For CW wind lidars like the ZX 300 series, the number of reconstructed wind vectors per interval depends on the number of configured measurement elevations as well as the amount of filtered data due to adverse atmospheric conditions. In addition, in practical applications, the random error visible as scatter on regression plots also depends on the distance between FLS and reference instrument and the overall uncertainty of FLS and reference instrument.

The ZX 300 wind lidar with current firmware performs VAD scans with a prism frequency of one rotation per second. Wind vectors are reconstructed based on line-of-sight data from one prism rotation. This study confirms results from field experiments that show that the measurement accuracy of FLS carrying a ZX 300 lidar is comparable to the performance of fixed lidar systems if the frequency of tilt motion is reasonably close to 0.4 Hz. For floating platforms with significantly different hydrodynamic properties that lead to different rotational frequencies and amplitudes, the expected measurement error can be approximated from Figures 7, 8, and 10. For lidar types with a different scanning strategy, similar simulations would need to be performed to determine the systematic measurement deviation. The choice of averaging, i.e., scalar or vector averaging of the reconstructed wind vectors, is of utmost importance for the quantification of the mean bias. All results in this paper are based on scalar averaging to imitate the processing of the ZX 300 lidar. If vector averaging would have been chosen instead, the mean bias values would have been independent of the frequency of motion and always negative. Translational motion would in this case not lead to any mean bias.

A useful continuation of this study would be to quantify the motion-induced measurement error for a real FLS measurement campaign. This can be done easily by replacing the generic periodic oscillations used as simulator input in this study by time series of measurement data for motion in all six DoF. The modelled wind shear profiles have to be replaced by the measured vertical profiles of mean wind speeds and directions. The resulting mean bias values can then be used to achieve compensation

of the effect of motion on mean wind speed estimates. Like all other motion-compensation methods, the performance of this
approach will be hard to assess because its effect is small compared to the uncertainties in a test setup.

Having confirmed that the systematic motion-induced bias on current FLS is low, it remains to be investigated in how far
lidar motion influences lidar internal data processing routines with regard to data filtering and cloud detection. These could add
a different dimension of uncertainty caused by motion.

**Appendix A:  Analytic modelling of buoy mean bias**

For the analytic approach we assume that the line-of-sight velocities $v_r(\theta')$ are given as a continuous function of the nominal
azimuth angle $\theta'$ which is a sound assumption given that the lidar performs 50 measurements per round. The horizontal wind
vector $\boldsymbol{U}_l = (U_l, V_l)$ is calculated from the these line-of-sight velocities according to Eqs. 2 and 3 by

$$U_l \sin\phi = B = \frac{1}{\pi} \int_0^{2\pi} v_r \cos\theta' d\theta' \tag{A1}$$

$$V_l \sin\phi = C = \frac{1}{\pi} \int_0^{2\pi} v_r \sin\theta' d\theta', \tag{A2}$$

where $\phi$ is the half-opening angle and the argument $\theta'$ of $v_r$ is understood. In this theoretical model we assume that the lidar is
capable of measuring the sign as well as the magnitude of the line-of-sight wind speed. This assumption explains many of the
small differences between theory and simulation seen in figures 6, 7 and 10. We now assume without loss of generality that the
wind is aligned with first axis $\boldsymbol{U} = (U, 0)$. Then $C/B$ is small and the length of the lidar estimated wind vector is

$$|\boldsymbol{U}_l| = \frac{B}{\sin\phi} \sqrt{1 + \left(\frac{C}{B}\right)^2}$$
$$\approx \frac{1}{\sin\phi} \left(B + \frac{1}{2}\frac{C^2}{B}\right) \tag{A3}$$

**A1  Pitch only**

The pitch angle $\varphi$ is defined as a harmonic variation as a function of time

$$\varphi = A\cos(\omega_\varphi(t - t_0)) \tag{A4}$$

where $A$ is the amplitude, $\omega_\varphi$ is oscillation frequency and $t_0$ is arbitrary initial time. The beam direction of a fixed lidar is

$$\boldsymbol{n} = \begin{pmatrix} \sin\phi \cos(\omega t - \phi_0) \\ \sin\phi \sin(\omega t - \phi_0) \\ \cos\phi \end{pmatrix} \tag{A5}$$

where the cyclic frequency is typically 1 Hz, so $\omega \approx 2\pi$ s$^{-1}$. The phase $\phi_0$ is random and may in some cases somewhat
surprisingly influence the results. The actual beam direction is obtained as the dot product between $\boldsymbol{n}$ and the rotation matrix

$\boldsymbol{M}$, which is given by

$$\boldsymbol{M} = \begin{pmatrix} \cos\varphi & 0 & \sin\varphi \\ 0 & 1 & 0 \\ -\sin\varphi & 0 & \cos\varphi \end{pmatrix} \tag{A6}$$

The beam direction of the floating lidar is thus $\boldsymbol{Mn}$ so the line-of-sight velocity is

$$
\begin{aligned}
v_r &= \boldsymbol{U} \cdot \boldsymbol{Mn} \\
&= U\left[\sin\phi\cos(\omega t - \phi_0)\cos(A\cos(\omega_\varphi(t-t_0))) + \cos\phi\sin(A\cos(\omega_\varphi(t-t_0)))\right]
\end{aligned} \tag{A7}
$$

or, if we define $\chi = \omega_\varphi/\omega$, $\theta' = \omega t - \phi_0$ and a random initial phase $\phi_r = \omega_\varphi t_0$, it can be written as

$$v_r = U\left[\sin\phi\cos\theta'\cos(A\cos(\chi(\theta'+\phi_0)-\phi_r)) + \cos\phi\sin(A\cos(\chi(\theta'+\phi_0)-\phi_r))\right] \tag{A8}$$

The ensemble average of $B$ is obtained by averaging over all random phases $\phi_r$:

$$\langle B \rangle = \frac{1}{2\pi^2}\iint\limits_0^{2\pi} v_r \cos\theta' \, d\theta' \, d\phi_r \tag{A9}$$

Inserting $v_r$ from (A8) and interchanging the order of integration one gets irrespective of the value of $\phi_0$ (so we do not need to average over that phase)

$$
\begin{aligned}
\langle B \rangle &= \frac{U}{\pi}\int_0^{2\pi} \sin\phi\cos^2\theta' J_0(A) d\theta' \\
&= U\sin\phi\, J_0(A)
\end{aligned} \tag{A10}
$$

where $J_0(A)$ is the Bessel function of the first kind. This means that the average wind component in the mean wind direction can be estimated as

$$U = \frac{\langle B \rangle}{\sin\phi\, J_0(A)} \tag{A11}$$

and where the bias correction $J_0(A)$ depends on the amplitude $A$ but not the non-dimensional frequency $\chi$.

Equation (A3) says that the random variations in the transverse wind speed contributes to the average of the length of the horizontal wind vector. The average of $C$ is zero but the variations around zero have to be calculated. The average of $C^2$ is calculated by multiplying the right hand side of (A2) with itself, substituting $\theta' \to \theta''$ in one of the integrals and converting the product into a double integral. In this calculation it is necessary to include the random initial phase $\phi_0$ and we have to average

over both $\phi_r$ and $\phi_0$ to get

$$\langle C^2 \rangle = \frac{1}{4\pi^4} \int\!\!\!\int\!\!\!\int\!\!\!\int_0^{2\pi} v_r(\theta')v_r(\theta'')\sin\theta'\sin\theta''d\theta'd\theta''d\phi_r d\phi_0$$

$$= \frac{U^2}{4\pi^4} \int\!\!\!\int\!\!\!\int\!\!\!\int_0^{2\pi} [\sin\phi\cos\theta'\cos(A\cos(\chi(\theta'+\phi_0)-\phi_r)) + \cos\phi\sin(A\cos(\chi(\theta'+\phi_0)-\phi_r))]$$

$$\times [\sin\phi\cos\theta''\cos(A\cos(\chi(\theta''+\phi_0)-\phi_r)) + \cos\phi\sin(A\cos(\chi(\theta''+\phi_0)-\phi_r))]$$

$$\times \sin\theta'\sin\theta''d\theta'd\theta''d\phi_r d\phi_0 \tag{A12}$$

The idea now is to assume $A$ is small so that the trigonometric functions containing $A$ can be approximated by its second order Taylor series as $\cos x \approx 1 + x^2/2$ and $\sin x \approx x$. It can be shown that using this expansion, the only term left after expanding the product between the two parentheses in (A12) is the product between the $\sin(A...)$ terms. Retaining terms of second order in $A$ one gets

$$\langle C^2 \rangle = \frac{U^2}{4\pi^4} \int\!\!\!\int\!\!\!\int\!\!\!\int_0^{2\pi} A^2\cos^2\phi\cos(\chi(\theta'+\phi_0)-\phi_r)\cos(\chi(\theta''+\phi_0)-\phi_r)\sin\theta'\sin\theta''d\phi_r d\phi_0 d\theta' d\theta'' \tag{A13}$$

where we have also changed the order of integration. It is convenient to make the following change of variables $\dot\theta = \theta' + \phi_0$ and $\ddot\theta = \theta'' + \phi_0$. Thereby (A13) becomes

$$\langle C^2 \rangle = \frac{A^2 U^2 \cos^2\phi}{4\pi^4} \int\!\!\!\int\!\!\!\int\!\!\!\int_0^{2\pi} \cos(\chi\dot\theta - \phi_r)\cos(\chi\ddot\theta - \phi_r)\sin(\dot\theta - \phi_0)\sin(\ddot\theta - \phi_0)d\phi_r d\phi_0 d\dot\theta d\ddot\theta \tag{A14}$$

Now the integration over $\phi_r$ and $\phi_0$ can be done separately over the two first and the two last terms in the integrand which results in

$$\langle C^2 \rangle = \frac{A^2 U^2 \cos^2\phi}{4\pi^2} \int\!\!\!\int_0^{2\pi} \cos(\chi(\dot\theta - \ddot\theta))\cos(\dot\theta - \ddot\theta)d\dot\theta d\ddot\theta$$

$$= \frac{A^2 U^2 \cos^2\phi}{\pi^2} \frac{(1+\chi^2)\sin^2(\pi\chi)}{(1-\chi^2)^2}, \tag{A15}$$

see also discussion after (A33). Averaging (A3),

$$\langle |\boldsymbol{U}_l| \rangle \approx \frac{1}{\sin\phi}\left(\langle B \rangle + \frac{1}{2}\frac{\langle C^2 \rangle}{\langle B \rangle}\right) \tag{A16}$$

and substituting (A10) for the average of $B$ and (A15) for the average of $C^2$ in this equation, we finally get

$$\langle |\boldsymbol{U}_l| \rangle \approx U\left[J_0(A) + \frac{A^2}{J_0(A)}\frac{\cot^2\phi}{2\pi^2}\frac{(1+\chi^2)\sin^2(\pi\chi)}{(1-\chi^2)^2}\right]. \tag{A17}$$

Figure 6 shows plots for pitch motion with three different amplitudes $A$.

## A2 Yaw only

The impact of harmonic yaw oscillations on the mean wind speed can be calculated in almost exactly the same way. Here the rotation matrix (A6) will become

$$\boldsymbol{M} = \begin{pmatrix} \cos\varphi & \sin\varphi & 0 \\ -\sin\varphi & \cos\varphi & 0 \\ 0 & 0 & 1 \end{pmatrix} \tag{A18}$$

and the only modification to $v_r$ in (A8) will be that in the second term $\cos\phi$ will be changed to $\sin\phi\sin\theta'$. Following the same steps as in section A1 one arrives to

$$\langle|\boldsymbol{U}_l|\rangle \approx U\left[J_0(A) + \frac{A^2}{J_0(A)}\frac{1}{\pi^2}\frac{(3\chi^4 - 12\chi^2 + 32)\sin^2(\pi\chi)}{8(\chi^3 - 4\chi)^2}\right] \quad . \tag{A19}$$

Figure 6 also shows plots for yaw motion with three different amplitudes $A$.

## A3 Pitch and shear

If the wind is not constant with height our results might change. Here we assume that the wind profile can be described by a power law profile with an exponent $\alpha$ (see Eq. 1). It is well known that the the curvature of the wind profile is important for the bias when averaging over a range of heights. However, here it turns out that the curvature is of small importance relative to the gradient. Anyway, we expand the wind profile to second order

$$U(z) \approx U_0\left(1 + \alpha\frac{\Delta z}{z_0} + \frac{\alpha(\alpha - 1)}{2}\left(\frac{\Delta z}{z_0}\right)^2\right) \tag{A20}$$

in order to see this small dependence on the second order in $\Delta z/z_0$. Using Eqs. A5 and A6, the relative height difference becomes

$$\frac{\Delta z}{z_0} = \frac{(\boldsymbol{Mn})_3}{n_3} - 1 = \cos(A\cos(\chi\theta' - \varphi_r)) - \cos\theta'\sin(A\cos(\chi\theta' - \varphi_r))\tan\phi - 1 \tag{A21}$$

utilizing the notation of Eq. A8. We now substitute Eq. A20 into Eq. A9 and expand all terms, integrate with respect to $\varphi_r$, and then with respect to $\theta'$. The result is

$$\frac{\langle B\rangle}{\sin\phi} = U_0\Big(J_0(A) - \alpha\left[J_0(A) - J_0(2A)\right]$$

$$+ \frac{\alpha(\alpha - 1)}{32}\left[J_0(A)(20 + 3\tan^2\phi) - 32J_0(2A) + 3J_0(3A)(4 - \tan^2\phi)\right]\Big) \tag{A22}$$

and the term with $\alpha(\alpha - 1)$ is typically insignificant relative to the term with $\alpha$ corresponding to the influence of the second and first derivative of the wind profile, respectively. When $\alpha = 0$ we are left with the result of Eq. A10. When $\alpha = 1$, that is $U \propto z$ the last term vanishes and $\langle B\rangle/\sin\phi = U_0 J_0(2A)$.

We now need to see if $C$ is affected by shear. If $A$ is small, then the relative height difference can be written as

$$\frac{\Delta z}{z_0} \approx -A\tan\phi\cos\theta'\cos(\chi\theta'-\varphi_r) - \frac{A^2}{2}\cos^2(\chi\theta'-\varphi_r) \tag{A23}$$

applying the same argumentation that led from Eq. A12 to Eq. A13. Using this it can be shown that the shear induced terms are third order in $A$ and are therefore dropped. The final result is thus Eq. A16 using Eq. A15 for $\langle C^2 \rangle$ and Eq. A22 for $\langle B \rangle$. Plots of the resulting bias for pitch motion with $A = 10°$ and $\alpha = 0\ldots0.15$ are presented in Figure 7.

## A4 Roll and shear

In a yaw only situation, the shear will not change the line-of-sight velocity because the measurement height remains constant. Neither the roll motion in a constant wind profile will alter the line-of-sight velocity because roll will not change the along wind component of the beam unit vector. Therefore we investigate the impact of shear combined with roll. The rotation matrix corresponding to roll motion is

$$\boldsymbol{M} = \begin{pmatrix} 1 & 0 & 0 \\ 0 & \cos\varphi & \sin\varphi \\ 0 & -\sin\varphi & \cos\varphi \end{pmatrix} \tag{A24}$$

where $\varphi$ is still given by Eq. A4 but $\varphi$ now means the roll angle. Since this matrix does not change the first component of the pointing vector $\boldsymbol{n}$ it is only the change in measurement height that will alter $v_r$. In parallel with Eq. A21 we can now calculate the relative height change of the focus position:

$$\frac{\Delta z}{z} = \frac{(\boldsymbol{Mn})_3}{n_3} - 1 = \cos(A\cos(\chi\theta'-\varphi_r)) - \sin\theta'\sin(A\cos(\chi\theta'-\varphi_r))\tan\phi - 1 \tag{A25}$$

Just using up to the first order in the expansion of the wind profile Eq. A20, the average of $B$ becomes

$$\langle B \rangle = \frac{1}{2\pi^2}\int\!\!\int_0^{2\pi} v_r\cos\theta'\,d\varphi_r d\theta' $$

$$= \frac{U_0\sin\phi}{2\pi^2}\int\!\!\int_0^{2\pi}\left(1+\alpha\frac{\Delta z}{z}\right)d\varphi_r\cos^2\theta'\,d\theta'. \tag{A26}$$

Now we substitute the relative height difference Eq. A25 and integrate first over $\varphi_r$ and then over $\theta'$ to get

$$\langle B \rangle = \frac{U_0\sin\phi}{\pi}\int_0^{2\pi}(1+\alpha(J_0(A)-1))\cos^2\theta'\,d\theta' $$

$$= U_0\sin\phi\,(1+\alpha(J_0(A)-1)). \tag{A27}$$

Taking into account the curvature of the wind profile, i.e. the second order term in Eq. A20, does not change the result significantly. That means the already small correction is changed by less than 10%. For completeness we give the expression

$$
\langle B \rangle = U_0 \sin\phi \left[ 1 + \alpha(J_0(A) - 1) + \right.
$$
$$
\left. \frac{\alpha(\alpha - 1)}{2} \left( 1 - 2J_0(A) + \frac{1 + J_0(2A)}{2} + \frac{1 - J_0(2A)}{8} \tan^2\phi \right) \right]. \tag{A28}
$$

To complete the analysis the impact of roll and shear on the average lidar speed we need to calculate $\langle C^2 \rangle$, according to

590 Eq. A16. Following the discussion after Eq. A24 and expanding $\Delta z/z$ to its first order Taylor series the line-of-sight velocity becomes

$$
v_r = \boldsymbol{U} \cdot \boldsymbol{M}\boldsymbol{n} = U(z)\sin\phi\cos\theta'
$$
$$
\approx U_0 \left( 1 + \alpha\frac{\Delta z}{z} \right) \sin\phi\cos\theta'. \tag{A29}
$$

Using this and Eq. A12, $\langle C^2 \rangle$ can be written as

$$
\langle C^2 \rangle = \frac{U_0^2 \sin^2\phi}{2\pi^3} \iiint\limits_0^{2\pi} \left( 1 + \alpha\frac{\Delta z'}{z} \right) \left( 1 + \alpha\frac{\Delta z''}{z} \right) d\phi_r \cos\theta' \cos\theta'' \sin\theta' \sin\theta'' d\theta' d\theta''. \tag{A30}
$$

Here the primes and double primes on $\Delta z$ correspond to primed or double primed $\theta$'s in Eq. A25 and for simplicity we assume $\phi_0 = 0$. We again use the expansion of the trigonometric functions in Eq. A25 assuming $A$ is small, so

$$
\frac{\Delta z'}{z} \approx \frac{A^2}{2} \cos(\chi\theta' - \varphi_r) - A\sin\theta' \tan\phi \cos(\chi\theta' - \varphi_r). \tag{A31}
$$

Expanding the two parentheses inside the triple integral in Eq. A30 gives four terms. The first three are independent of $\theta'$ and

600 $\theta''$ when integrated over $\varphi_r$, and thus, the integrals over $\theta'$ and $\theta''$ of those terms are null. We are left with

$$
\langle C^2 \rangle = \frac{\alpha^2 U_0^2 \sin^2\phi}{2\pi^3} \iiint\limits_0^{2\pi} \frac{\Delta z'}{z} \frac{\Delta z''}{z} d\phi_r \cos\theta' \cos\theta'' \sin\theta' \sin\theta'' d\theta' d\theta''. \tag{A32}
$$

We now substitute Eq. A31 into Eq. A32 and retain only terms of up to second order in $A$. The resulting expression is

$$
\langle C^2 \rangle = \frac{\alpha^2 A^2 U_0^2 \sin^2\phi \tan^2\phi}{2\pi^3} \iiint\limits_0^{2\pi} \cos(\chi\theta' - \varphi_r)\cos(\chi\theta'' - \varphi_r) d\phi_r \cos\theta' \cos\theta'' \sin^2\theta' \sin^2\theta'' d\theta' d\theta''. \tag{A33}
$$

Because of the general identity $\int_0^{2\pi} \cos(a - t)\cos(b - t)dt = \pi\cos(a - b)$, the integral over $\varphi_r$ of the two first cosine terms in

the above equation is $\pi\cos(\chi(\theta' - \theta''))$ so the final expression becomes

$$
\langle C^2 \rangle = \frac{\alpha^2 A^2 U_0^2 \sin^2\phi \tan^2\phi}{2\pi^2} \iint\limits_0^{2\pi} \cos(\chi(\theta' - \theta'')) \cos\theta' \cos\theta'' \sin^2\theta' \sin^2\theta'' d\theta' d\theta''
$$
$$
= \frac{\alpha^2 A^2 U_0^2 \sin^2\phi \tan^2\phi}{2\pi^2} \frac{16\chi^2 \sin^2(\pi\chi)}{(\chi^2 - 1)(\chi^2 - 9)}. \tag{A34}
$$

The final bias depends according to (A15) both on $\langle B \rangle$ and $\langle C^2 \rangle$. However, the term involving $\langle C^2 \rangle$ is second order in $\alpha$ in contrast to the bias due to $\langle B \rangle$ which is first order in $\alpha$, see (A27). Since the relevant values of $\alpha$ are small, typically around 1/7

or less, and the other terms entering into the expression for $\langle C^2 \rangle$ in (A34) are also limited in magnitude for relevant parameters, $\langle C^2 \rangle$ can be safely ignored, and the final expression for the mean speed bias due to roll and shear is

$$\langle |\boldsymbol{U}_l| \rangle \approx \frac{\langle B \rangle}{\sin \phi} = U_0 \left(1 + \alpha(J_0(A) - 1)\right) \tag{A35}$$

where we have used Eq. A27. Plots of the resulting bias for roll motion with $A = 10°$ and $\alpha = 0 \ldots 0.15$ are presented in Figure 8.

**A5    Translational motions**

For translational motions the line-of-sight speed is in the case of no shear which is the only case considered in this section $v_r = (\boldsymbol{U} + \boldsymbol{v}) \cdot \boldsymbol{n}$, where the mean wind speed is $\boldsymbol{U} = (U, 0, 0)$, the translational motion is $\boldsymbol{v} = \hat{\boldsymbol{v}} \cos(\omega_t(t - t_0))$ with the amplitude given by

$$\hat{\boldsymbol{v}} = \hat{v} \times \begin{cases} (1,0,0) & \text{surge} \\ (0,1,0) & \text{sway} \\ (0,0,1) & \text{heave} \end{cases} \tag{A36}$$

for the motion in the three spatial directions. The unit vector of the beam direction $\boldsymbol{n}$ is still given by (A5). Substituting $\omega t - \phi_0$ with $\theta'$ and $\chi \equiv \omega_t / \omega$ the line-of-sight velocity can be written as

$$v_r = U \sin\phi \cos\theta' + \hat{v} \cos(\chi(\theta' + \phi_0) - \phi_r) \begin{pmatrix} \sin\phi\cos\theta' \\ \sin\phi\sin\theta' \\ \cos\phi \end{pmatrix} \tag{A37}$$

where the three terms in the vector correspond to the surge, sway and heave motions, respectively. Using the definitions of $B$ (A1) and $C$ (A2) and taking the average by integrating over $\phi_r$ it is easily seen that the bias on both $B$ and $C$ is zero. So if

one did vector averaging there would be no bias due to translational motions. However, since vector averaging is the current standard we have to use (A16) and calculate $\langle C^2 \rangle$ which is also equal to the bias:

$$\langle C^2 \rangle = \left\langle \frac{1}{\pi^2} \int\!\!\!\int_0^{2\pi} v_r(\theta') v_r(\theta'') \sin\theta' \sin\theta'' d\theta' d\theta'' \right\rangle \tag{A38}$$

where $\langle\rangle$ means averaging over $\phi_0$ and $\phi_r$. Doing the same substitutions that led to (A14) and integrating over $\phi_r$ one gets

$$\langle C^2 \rangle = \frac{\hat{v}^2}{4\pi^4} \iiiint \cos(\chi\dot{\theta} - \phi_r)\cos(\chi\ddot{\theta} - \phi_r) \begin{pmatrix} \sin^2\phi\cos(\dot{\theta} - \phi_0)\cos(\ddot{\theta} - \phi_0)\sin(\dot{\theta} - \phi_0)\sin(\ddot{\theta} - \phi_0) \\ \sin^2\phi\sin^2(\dot{\theta} - \phi_0)\sin^2(\ddot{\theta} - \phi_0) \\ \cos^2\phi\sin(\dot{\theta} - \phi_0)\sin(\ddot{\theta} - \phi_0) \end{pmatrix} d\dot{\theta}d\ddot{\theta}d\phi_0 d\phi_r$$

$$= \frac{\hat{v}^2}{4\pi^3} \iiint \cos(\chi(\dot{\theta} - \ddot{\theta})) \begin{pmatrix} \sin^2\phi\ \cos(\dot{\theta} - \phi_0)\cos(\ddot{\theta} - \phi_0)\sin(\dot{\theta} - \phi_0)\sin(\ddot{\theta} - \phi_0) \\ \sin^2\phi\ \sin^2(\dot{\theta} - \phi_0)\sin^2(\ddot{\theta} - \phi_0) \\ \cos^2\phi\ \sin(\dot{\theta} - \phi_0)\sin(\ddot{\theta} - \phi_0) \end{pmatrix} d\dot{\theta}d\ddot{\theta}d\phi_0 \tag{A39}$$

where all integrals are from $0$ to $2\pi$. Now we do the integration over $\phi_0$

$$\langle C^2 \rangle = \frac{\hat{v}^2}{2\pi^2} \iint \cos(\chi(\dot{\theta} - \ddot{\theta})) \begin{pmatrix} \sin^2\phi\ \frac{1}{8}\cos(2(\dot{\theta} - \ddot{\theta})) \\ \sin^2\phi\ \frac{1}{8}(2 + \cos(2(\dot{\theta} - \ddot{\theta}))) \\ \cos^2\phi\ \frac{1}{2}\cos(\dot{\theta} - \ddot{\theta}) \end{pmatrix} d\dot{\theta}d\ddot{\theta} \tag{A40}$$

and after performing the double integral over $\dot{\theta}$ and $\ddot{\theta}$ we finally arrive at

$$\langle C^2 \rangle = \frac{\hat{v}^2}{2\pi^2}\sin^2(\pi\chi) \begin{pmatrix} \sin^2\phi\ \frac{4+\chi^2}{2(4-\chi^2)^2} \\ \sin^2\phi\ \frac{32-12\chi^2+3\chi^4}{2\chi^2(4-\chi^2)^2} \\ \cos^2\phi\ \frac{2(1+\chi^2)}{(1-\chi^2)^2} \end{pmatrix} \quad . \tag{A41}$$

The resulting biases for the three translational DoF are presented in Figure 10.

*Author contributions.* "FK designed the study, performed the numerical simulations, and prepared the manuscript. JM developed the analytic model and reviewed the manuscript. All authors analyzed the results."

*Competing interests.* Fugro Norway AS is the manufacturer of the SEAWATCH Wind LiDAR Buoy. The author FK is an employee of Fugro Norway AS.

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
