# Peer review of "Quantification of Motion-Induced Measurement Error on Floating Lidar Systems"

_Atmospheric Measurement Techniques, 2022_

## Author Comment (AC1)

**Response to comment by Anonymous Referee #1**

*Please find our response to Anonymous Referee #1 in blue italic font below their comments.*

**Referee comment:**

The paper addresses the quantification of the error, induced by the motion of a floating lidar and explains the logic behind the low error values obtained, with such a lidar, from a field trial.

Both the 'simulator' and the 'analytical' methods reveal that the pitch motion (tilt motion // in the wind direction) is the predominant one and capable to produce a systematic bias error. The developed analytic model is of significant value and can be applied to PW-lidars too.

The main conclusion of the paper is that the expected motion-induced wind speed error, for the wave motions recorded during an (older) field campaign, is lower than the uncertainty of the ref. wind sensors (cup).

A general comment  for the paper (or suggestion for future work), is that both models are extremely difficult to be verified by offshore field campaigns. The expected errors are small and some influencing factors, like: i) the separating distance fixed-MM/lidar to floating lidar, ii) the fixed-MM blockage effect and iii) the different probe volumes (fixed lidar is usually 10m higher than the floating one), can be sources of deviations. Instead, a campaign similar to the mentioned Hellevang and Reuder (2013) would be ideal to verify the models accuracy.

*We thank Anonymous Referee #1 for their review of our discussion paper. We agree with the comment that an experimental verification of our findings by means of an offshore trial is practically impossible for the reasons mentioned. The alternative of using a forced-motion experiment as in Hellevang and Reuder (2013) appears more realistic. Still, while in such a set-up the motion can be controlled, environmental conditions like the vertical profile of wind speeds and directions will change uncontrollably. This would introduce variability into the measurements and the resulting uncertainty might be too high for a conclusive comparison of measurements and the analytical solution which assumes steady conditions over a long time. The simulations though might be verified after adapting them to more variable conditions.*

---

## Author Comment (AC2)

**Response to comment by Anonymous Referee #2**

*Please find our response to Anonymous Referee #2 in blue italic font below their comments.*

**General Comments**

Floating Lidar Systems (FLS) are being accepted in the wind energy industry as a trustable mean wind measurement source. The motion influence on mean HWS measurement by FLS has not been quantified as it is usually lower than the sensor uncertainties. This paper provides a convenient method based on a FLS motion simulator to estimate the mean HWS measurement error, which is validated by analytical methods. The paper contents are more focused towards a theoretical understanding than towards an experimental implementation. A complete review of the state of the art is given in the introduction, providing a solid background to motivate the study.

A thorough analysis of the error as a function of different motional parameters and the wind shear is given. Further, two motional case examples are studied: "normal" and "strong" wave motion. The estimated HWS mean measurement error figures obtained are of an order of magnitude lower than those observed experimentally, and they could not be validated experimentally by the authors.

The analytical formulation of the Appendix is nicely formulated and its description permits the reader to follow its derivation in a structured way. Moreover, its description as a function of the motional scenario allows the reader to better understand the HWS mean bias sources. However, some steps are unrelated to or isolated from the manuscript and more context is required.

Although the paper contents are aligned with the state of the art and give original results, the presentation of methods and the discussion of results are somewhat misleading and/or unclear, and could be improved (please see SPECIFIC COMMENTS below). This considered as well as the long list of specific comments below I'm recommending a major revision. The manuscript looks promising and it will definitely improve by implementing the proposed changes below.

*We thank Anonymous Referee #2 for their excellent review of our discussion paper and appreciate the constructive nature of the feedback given. It was essential to finding and removing significant errors present in the discussion paper. The revised version of the paper has been improved according to the reviewer's recommended changes. This accounts not only for many minor points but also for two major corrections that we would like to highlight here.*

*First, the influence of translational motion on mean wind speed estimates has been included. The reason for disregarding it in the first version of the paper was a programming mistake in the simulator that we found after doubts have been raised by the referee in their report (see below).*

*Second, the interaction of the first phase angle and the motion phase angle in the simulator and in the analytical solution was not considered correctly. We introduced the phase offset angle between these two angles in order to reduce complexity for better understanding of VAD sampling under the influence of motion. Unfortunately, the assumptions we made hold only for cases of oscillatory motion with very low frequency and frequencies in resonance*

*with the prism rotation. For all other frequencies our results were erroneous. This has now been fixed. Although these two changes in simulator and theory lead to different results for the biases of the test cases, the conclusion that motion-induced errors on FLS are small remains valid.*

**Specific Comments**

ALL OVER THE MANUSCRIPT

Please CONTEXTUALISE: There are some equations and assumptions which are uncontextualized and require some background explanation for the inexpert reader to be understood. As an example, the VAD algorithm is briefly introduced (figure-of-eight fitting), and expressions derived from its computation procedure (such as Eqs. A1 and A2) are presented without any context and assumed to be known beforehand. This can be found in the simulator explanation (e.g., the law wind profile and the lidar scanning procedure) in Sect. 2.1 and the equations of Sect. 2.5 and the Appendix, please revise.

*We have now added context to many passages of the text. The VAD processing routine, for example, is introduced in the main text and where it is helpful in the appendix cross-references are set. Also, the power-law profile is now introduced where we first refer to it and referenced later.*

Sect. 2 Materials and Methods

Errors discussed in the paper must be defined formally. It is hard to discern between single-scan and 10-min errors. Is the 10-min error derived from the average of 600 simulated scans? Are the errors in sections from 2.4 to 2.9 defined for 10-min observations?, or only for a scan? The difference between systematic and random errors is also unclear. Please notate variables accordingly.

*In the revised version we are stricter in referring to the mean bias (MB) were the average of many single-scan errors is meant. Otherwise, we simply use the term relative wind speed error. This has been revised in the text, figures, and captions. We added clarification that only systematic errors are considered in the study by defining MB. Furthermore, in the discussion of the results we describe why the analysis of the random errors appears not useful for this study.*

2.1 Lidar simulator

A more detailed description of the lidar measurement procedure and how the motion is emulated in the algorithm is required. This could be carried out by means of a block diagram, or a list, explaining more in depth the lidar scanning procedure (VAD algorithm) and the rotational motion influence on the lidar pointing direction. Consider adding equations to define the wind vector, power law wind profile, and the real lidar pointing direction. Some simulator steps added later in the manuscript are not explained, e.g., the influence of pitch motion phase.

*The description of the lidar simulator is extended in the revised version of the paper. For a more detailed description of the lidar measurement procedure we refer to Kelberlau and Mann (2019) and for the derivation of the real lidar pointing direction we refer to the section*

*of Kelberlau et al. (2020) in which we describe it. is We also added equations for the power-law wind profile, the VAD wind vector reconstruction and the mean bias.*

Regarding the first phase angle consideration, are each of the *100 simulated test cases* carried out considering the same phase offset for all *600 lidar scans*? In other words, is the initial phase angle randomness considered individually for each of the 600 scans, or a single value the same for the whole set? Please clarify. Consider providing an equation for it.

*This was indeed unclear in the discussion paper. We found that the high number of iterations with different phase angles was not adding any other results than choosing lower numbers. Therefore, we reduced the scan time from 600 seconds to 10 one-second scan with varying lidar phase angles and the number of motion phase angles from 100 to 20. The averaging procedure is now better described by explaining that: "Ten seconds of scan time with ten different first phase angles times twenty different phase offsets of motion will lead to 200 reconstructed wind vectors."*

2.4 Lidar motion in six degrees of freedom

The translational motion is disregarded by assuming that the displacement around a fixed point for FLSs anchored to the seabed is null in average. Is the average translational velocity equal to exactly 0 m/s for 10-min periods? Since the HWS bias figures found show small values as well, does not the translational motion contribute to it? Further information or references should be provided to motivate this assumption.

*We think that the assumption of translational motion being zero on average makes sense for seabed anchored FLS. The average translational velocity during one 10-min period is only exactly 0 m/s if the FLS is located at the exact same location in the beginning and the end of the averaging interval. This is not necessarily the case for each single interval, but it is correct for the average of all possible intervals.*

*Nonetheless, it was wrong to disregard translational motion. It was a glitch to assume that oscillatory translational motion does not contribute to the mean bias. When reconstructed wind vectors are scalar-averaged, translational motion leads to a positive bias. We therefore included it in the revised version of the paper.*

2.5 Pitch motion with no wind shear

Both the lidar first phase angle and the pitch motion phase nominal values have influence on the HWS bias in a scan. Is this considered for the results of the paper (Fig. 6 for example)? How? Or is only the phase offset considered for the 1-Hz case (Fig. 5)? Please clarify.

*The referee raises an important issue here. Both phase angles (lidar prism and motion) can influence the mean bias of HWS and must be considered for the results of the paper. For the description of the 0 and 1-Hz case it is possible to simplify the situation by combining both phase angles to only one variable, the phase offset angle. In all other frequency cases both phase angles must be kept independent.*

*Figs. 4 & 5 are therefore correct, but Figs. 6 & 7 had to be changed. For the new figures the simulator was modified to actually behave like described in the text. The analytic solution*

*now considers not only the arbitrary initial time $t_0$ but also the random lidar prism phase angle $\varphi_0$. Also, the test case results have changed.*

3.1 Tilt Frequency

Please justify that the SWLB is restricted to a narrow band, i.e., one narrow spectrum peak. Is Fig. 9 just an example case, or most of the scenarios are equivalent? Please justify this assumption or give a reference.

*The example of Fig. 9 is representative for all operating conditions of the FLS type. The dominant tilt frequency can be understood as the natural frequency of a mass-spring(-damper) model in the tilt DoF. Although the hydrodynamic added mass is a function of amplitude and frequency of motion, we experience the tilt frequency range to be narrow. This is now explained in the revised version of the paper.*

4 Discussion and conclusions

The experimental results obtained are not validated experimentally as the theoretical error is too small in comparison to the instrument uncertainties. Please, could the authors suggest an approach to validate these results? How these results compare with similar ones (if any) in the state of the art?

*The simulation results are calculated completely independent of the derivation of the equations for the analytical model. The comparison of data from both sources for motion in all degrees-of freedom with and without influence of wind shear serves as a cross validation. The good match between the two gives credibility to the simulator results. Experimental validation is probably not possible because of the uncertainty of reference instruments and trial setups. This lack of possibilities for experimental quantification of the measurement bias is motivating its theoretical estimation.*

Appendix

Please, provide a more accurate contextualization of the expressions derived in the Appendix: see technical corrections below. For example, explain either in the appendix or in the manuscript (and then cross-reference) the origin of Eqs. A1 and A2. Consider to provide more crossed references with respect to the manuscript.

*We provided more cross-references between appendix and the main text.*

**Technical Corrections**

Line 23: Reference needed for the wind speed - production relationship.
*Reference added to Heier, 2014*
Line 56: Rephrase to "10-min mean wind velocity by a FLS."
*Done*
Line 58: Rephrase to "wind lidar, taking as a reference the ZX 300M…".
*Done*
Line 58: "Computer simulations are validated by means of an analytic model."

*Done*
Line 61: "The bias is quantified for the SWLB by Fugro (Trondheim, Norway) under "normal" and "strong" wave conditions."
*Done*
Line 64: "with and without".
*Thanks, done*
Line 70: Remove line break.
*Done*
Line 74: Include the lidar scan time.
*We included "The resulting set of synthetic lidar data is then used to reconstruct wind vectors. Each of them is based on data of 50 samples representing one second scan time and one full prism rotation."*
Line 76: Please explain the vector transformations as well as the real azimuth and elevation angle derivation procedure.
*We rephrased and added in which section of Kelberlau et al. (2020) the requested explanation is found. The angle derivation is long and unhandy. We therefore decided to not repeat it here.*
Line 80: Mention the VAD algorithm.
*We included a description of the VAD algorithm.*
Line 81: Please include a sentence explaining how the mean bias is derived.
*We included equation (4) at the end of the paragraph.*
Line 88: Please, use the acronym everywhere or do not use it.
*The now use the acronym wherever suitable.*
Table 1 is not needed. It can be embedded in the text.
*Table 1 is removed now, and the information is embedded in the text.*
Line 96:  subsections from 2.4 to 2.9 as subsubsections of subsection 2.3.
*We created a new section 3 with subsections 3.1 to 3.7 for these subsections.*
Line 114: Add reference to justify the wind direction disregarding.
*We don't know of any reference that would support the statement. Instead, we now support our statement by writing that "Motion which is not aligned with the wind direction or being perpendicular to it can be decomposed into a linear combination of its surge/roll and sway/pitch components of motion."*
Line 116: Add reference to justify the assumption of no translational motion influence.
*Translational motion is now included as a source of mean bias (see above).*
Line 130: Remove the period after "important".
*Done*
Line 135: How Eq. 2 is reached?
*We added an explanation of how the equation is reached.*
Line 141: Rephrase to "… for static tilt (fp=0)."
*Done*
Please homogenize the font size in all figures of the manuscript.
*We will homogenize the font size in all figures in the context of final copy-editing when we know which figures will span the entire width of a page and which will be in one-column format.*
Figure 2 caption: Please define in the text upwind and downwind.
*We clarified in the text that upwind is "the direction from where the wind blows".*
Figure 3: Please add "a)", "b)", etc. labels to each of the figure panels. "Projection", "Top view", etc. labels must be included in the caption. Same for the rest of figures.
*"(a)", "(b)" has been added to the figure panels according to the journal standard.*
Lines 187-189: Please add a reference or clarification with regards to the scalar averaging and

vector averaging influence on the bias.

*We added clarification regarding the effect of scalar vs. vector averaging here and at several other relevant sections.*

Line 198: Please rephrase to "This can be explained by the approximation of A by means of the second order Taylor's expansion (see Eqs. A12-A13). Expanding A to a higher order would probably eliminate these small deviations due to approximation."

*We followed this suggestion to rephrase.*

2.6 Roll motion with no wind shear: This section could be omitted and merged into 2.5.

*Yes, it could be merged but we prefer to mention pitch, roll, and yaw DoF each in their own subsection.*

2.7 Yaw motion: This section could also be merged into 2.5.

*See previous response*

Line 212: Please add a reference.

*We added Elkinton et al. (2006) as reference.*

Line 216: Please specify what "results" refer to. Is it 10-min mean HWS bias or similar?

*We specified that we refer to "motion-induced measurement error on 10-min averages of horizontal wind velocity, i.e., the mean bias…".*

Line 248: Please provide more information about the measurement campaign from which the measurement data is used. Exact location, measurement time, etc.

*We provided more information about the origin of the used motion data "Figure 9 shows the single-sided power spectrum of IMU-measured tilt motion data measured by SWLB unit 056 in the period from 14:00 until 16:00 UTC on 12th November 2021 close to the town of Titran off the coast of Frøya, Norway."*

Line 253: Please provide the exact time of the measurement.

*See previous comment*

Line 253: Please give more information on the bin average procedure, e.g., number of averaged bins.

*We added "The frequency bins have a width of 0.01Hz each."*

Line 253: Please specify what "tilt" refers to. It is not clear if it is only pitch tilt or a combination of roll and pitch.

*We specified: "The measurement data presented here is tilt in one of the buoy's local coordinate system axes. For an axis-symmetric FLS like the SWLB the dominating tilt frequency is identical for pitch, roll, and their combination. It is therefore unnecessary to rotate the coordinate system of the motion data into a particular direction for this analysis."*

Line 263: Please provide a reference for significant wave height.

*It appears that Sverdrup and Munk (1947) introduced the definition of significant waves, so we added a reference.*

Line 264: Please provide exact location and campaign dates.

*The location is given by naming the East Anglia One meteorological mast. We provided the campaign dates.*

Line 273: Rephrase to "The mean tilt amplitude is defined here as the average of the local maxima of the rectified tilt time series". Does tilt refer to pitch?

*We rephrased according to the reviewer's suggestion. We also clarified that "Here, tilt refers to the quadratic sum of the rotation angles around both horizontal axes".*

Figure 10: Markers for "normal" and "strong" cases under study could help the reader.

*We added vertical lines marking the "normal" and "strong" cases.*

Line 281: Please provide reference.

*We added Elkinton et al. (2006) as reference.*

Line 297: Please add a coma after "UK".

*Done*

Line 310: Please refer to the particular experimental campaign.

*Instead of repeating the details of the experimental campaign, we refer back to previous section 4.*

Line 316: Please provide a reference or give typical mean HWS measurement bias figures.

*We added that "typical uncertainties [are] around 2%".*

Line 318: Please provide a reference to justify "the sensitivity of measurement error of SWLB to motion and sea-state parameters is insignificant".

*To the best of our knowledge no FLS provider (including Fugro) has published their classification reports. We added that reference can be shared with the interested reader on request.*

Line 323: What does random error refer to? What is the difference between systematic bias and random error? Please clarify in the text.

*We restructured the paragraph and added information and a cross-reference to be clearer.*

Line 345: Please clarify the origin of equations A1 and A2. Relate to VAD algorithm.

*We added a cross reference to the VAD equations (Eqs. 2&3).*

Equation A2: Please add a coma after the equation.

*Done*

Line 353: Please rephrase to "The pitch angle Ï• is defined as a harmonic variation as a function of time".

*Done*

Line 357: Please rephrase to "The actual beam direction is obtained as the dot product between **n** and the rotation matrix **M**, which is given by"

*Done*

Line 382: Please rephrase "can be expanded to second order" to "can be approximated by its second order Taylor series"

*Done*

Line 403: Please change ", see Eq. 4" into "(see Eq. 4)".

*Done*

Line 414: "we are left with the result **of** Eq. A10".

*Done*

Equation after line 430: There is no equation number. Period after the equation required.

*Equation number and period added*

Line 436: Please refine the sentence.

*We tried to improve the sentence.*

Equation A26: Please add a period after the equation.

*Done*

Line 440: Rephrase to "To complete the analysis of the impact of roll and shear on the average lidar speed we need to calculate C2, according to Eq. A15".

*We rephrased accordingly.*

Line 441: Change "expanding only to first order in âˆ†z/z" into "and expanding âˆ†z/z to its first order Taylor series,".

*Changed according to suggestion.*

Equation A27: Please add a period after the equation. Add punctuation to the remaining equations.

*We added punctuation.*

Line 444: Please remove "equation".

*OK*

Line 449: Please rephrase "The first three are, when integrated over $\varphi r$, independent of $\theta'$ and $\theta''$ so the integrals over $\theta'$ and $\theta''$ of those terms give zero. We are left with" to "The first three are independent of $\theta'$ and $\theta''$ when integrated over $\varphi r$, and thus, the integrals over $\theta'$

and θ ″ of those terms are null, leading to".

*We rephrased the sentence.*

Line 452: Please rephrase to "We now substitute Eq. A29 into Eq. A30 and retain only terms of up the second order in A."

*Done*

Line 454: Please rephrase the sentence, it is hard to understand.

*For better understanding we rephrased the sentence to "Because of the general identity $\int_0^{2\pi} \cos(a-t)\cos(b-t)dt = \pi \cos(a-b)$, the integral over $\varphi r$ of the two first cosine terms in the above equation is $\pi \cos(\chi(\theta' - \theta''))$ so the final expression become"*

Line 456-459: Please rephrase. Its meaning is unclear.

*We rephrased for clarity: "The final bias depends according to (A15) both on ⟨B⟩ and ⟨C2⟩. However, the term involving ⟨C2⟩ is second order in α in contrast to the bias due to ⟨B⟩ which is first order in α, see (A27). Since the relevant values of α are small, typically around 1/7 or less, and the other terms entering into the expression for ⟨C2⟩ in (A34) are also limited in magnitude for relevant parameters, ⟨C2⟩ can be safely ignored, and the final expression for the mean speed bias due to roll and shear is"*